# Optineurin is an adaptor protein for ubiquitinated substrates in Golgi membrane-associated degradation

Yoichi Nibe-Shirakihara [1] ✉, Shinya Honda[1], Satoko Arakawa [1], Satoru Torii [1], Hajime Tajima Sakurai[1], Hirofumi Yamaguchi[1], Shigeru Oshima [2], Ryuichi Okamoto [2], Michael Lazarou [3,4,5], Hideshi Kawakami [6] & Shigeomi Shimizu [1] ✉

Golgi membrane-associated degradation (GOMED) is a process that leading to the degradation of proteins that have passed through the *trans*-Golgi membranes upon Golgi stress. GOMED is morphologically similar to autophagy, but the substrates degraded are different, and they thus have different biological roles. Although the substrate recognition mechanism of autophagy has been clarified in detail, that of GOMED is completely unknown. Here we report that GOMED degrades its substrate proteins selectively via optineurin (OPTN), as we found that the degradation of GOMED substrates is s`uppressed by the loss of OPTN. OPTN binds to K33 polyubiquitin-tagged proteins that have passed through the Golgi, which are then incorporated into GOMED structures for eventual degradation. In vivo, GOMED is known to be involved in the removal of mitochondria from erythrocytes, and in Optn-deficient mice, mitochondria are not degraded by GOMED, resulting in the appearance of erythrocytes containing mitochondria. These findings provide insight into the substrate recognition mechanism of GOMED.

Several intracellular proteolysis mechanisms have been identified to date, such as autophagy and Golgi membrane-associated degradation (GOMED, also known as alternative autophagy or Rab9-dependeent autophagy)[1–3]. The role of autophagy is to degrade unnecessary and harmful intracellular substances (such as damaged lysosomes, mitochondria, abnormal proteins, and intracellular parasites), and to maintain cellular homeostasis. Initially, autophagy was considered to be a non-selective degradation system that randomly engulfs cytoplasmic components. However, when degrading unnecessary and harmful substances, autophagy can selectively engulf them, in a process known as "selective autophagy". GOMED is another intracellular proteolysis mechanism, which mainly degrades proteins that pass through the Golgi apparatus, such as secreted proteins and plasma membrane proteins[4–6]. GOMED is morphologically similar to autophagy, as both GOMED and autophagy engulf substrates in double-membrane structures, namely autophagosomes, and degrade them by fusion with lysosomes[4,6]. These two proteolysis mechanisms are also similar in that they are phylogenetically conserved from yeast to mammals[6]. However, they operate in different contexts, have different cellular functions, and degrade different cargos[6,7]. Furthermore, autophagic membranes originate from the endoplasmic reticulum (ER) membrane, particularly mitochondria-associated ER

[1]Department of Pathological Cell Biology, Advanced Research Initiative, Institute of Integrated Research, Institute of Science Tokyo, 2-3-10 Kanda Surugadai, Chiyoda-ku, Tokyo, Japan. [2]Department of Gastroenterology and Hepatology, Graduate School, Institute of Science Tokyo, 1-5-45 Yushima, Bunkyo-ku, Tokyo, Japan. [3]Department of Biochemistry and Molecular Biology, Biomedicine Discovery Institute, Monash University, Melbourne, Australia. [4]Walter and Eliza Hall Institute of Medical Research, Parkville, Victoria, Australia. [5]Department of Medical Biology, University of Melbourne, Melbourne, Victoria, Australia. [6]Department of Epidemiology, Research Institute for Radiation Biology and Medicine, Hiroshima University, Hiroshima, Japan. ✉e-mail: ynibe.pcb@tmd.ac.jp; shimizu.pcb@mri.tmd.ac.jp

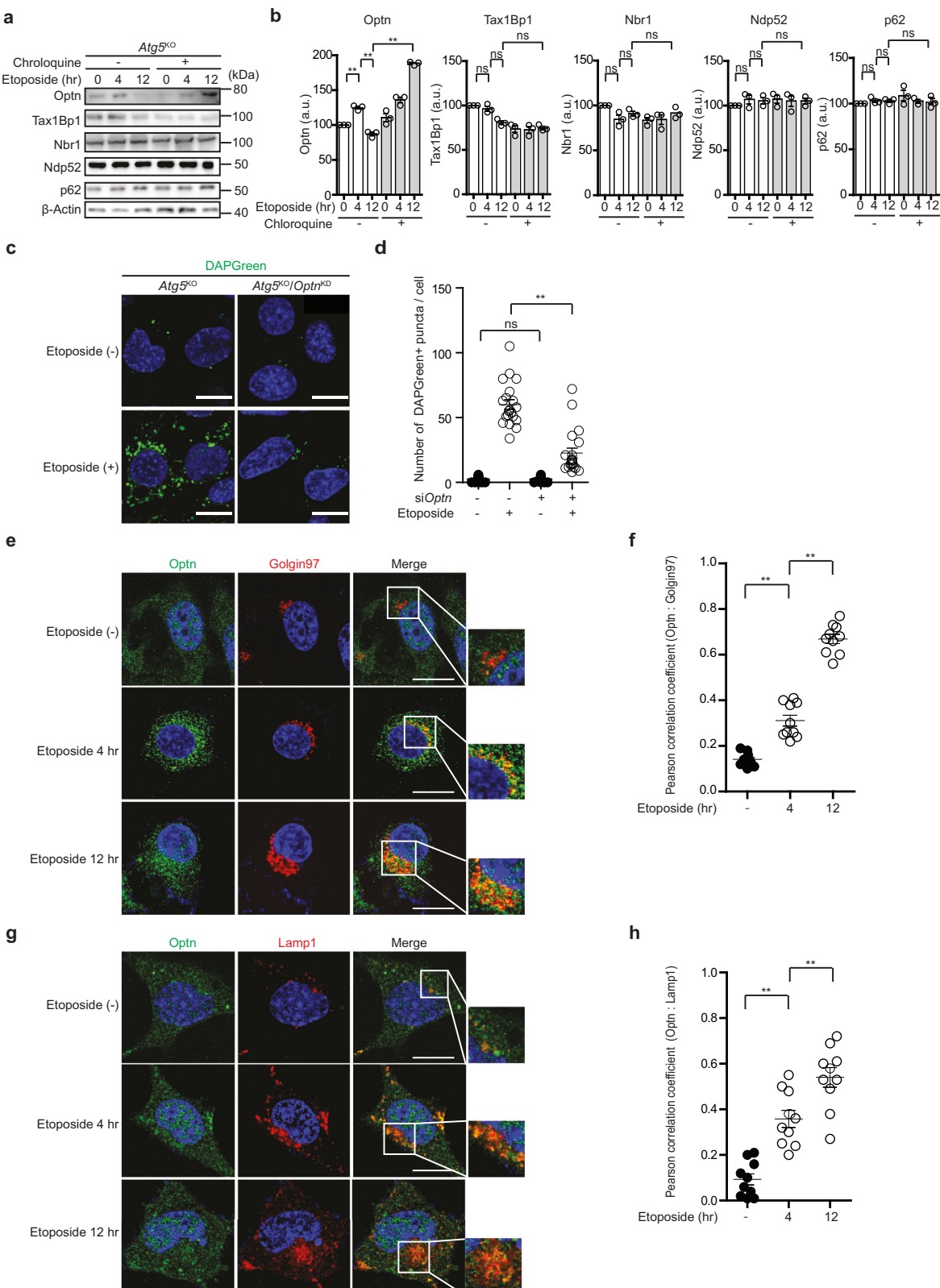

membranes[8,9], whereas GOMED membranes are derived from *trans*-Golgi membranes[4,6]. Regarding the molecules involved, autophagy is driven by functional complexes containing Atg9, Wipi1/2, the Atg5-conjugation system, and the LC3-conjugation system[1,10], but none of these molecules are required for GOMED[4]. GOMED requires different molecules, namely, Wipi3/4 and Rab9, which have minimal function in autophagy[4,11]. Regarding their physiological functions, GOMED

functions in cell maintenance via its degradation of ceruloplasmin in neurons, and in mitochondrial removal in the final step of mitochondrial degradation in erythrocytes[11–13], but autophagy is not involved in these phenomena.

Thus, the molecular dynamics and functions of GOMED are being clarified, but how GOMED recognizes and degrades its substrates has not been elucidated to date. In selective autophagy, ubiquitin (Ub) is

**Fig. 1 | Involvement of Optn in etoposide-induced GOMED. a** The dynamics of autophagy adaptor proteins after genotoxic stress. *Atg5*[KO] MEFs were treated with etoposide (10 μM) in the presence or absence of chloroquine (100 μM). At the indicated times, cells were lysed, and the expression of each protein was analyzed by western blotting. ß-actin was used as a loading control. **b** Semiquantitative analysis of protein expression in the experiment in (**a**). Data are shown as the mean ± SE (*n* = 3). **c, d** Involvement of Optn in etoposide-induced GOMED. *Optn* was silenced in *Atg5*[KO] MEFs, and the indicated MEFs were treated with etoposide (10 μM) for 12 hr. Then, the extent of GOMED was analyzed using DAPGreen. Nuclei were counterstained with Hoechst 33342. Representative images are shown in (**c**) Bars = 10 μm. In **d** the number of DAPGreen puncta per cell was calculated. Data are shown as the mean ± SE (*n* = 20 cells, experiments were performed two times). **e, f** Etoposide-induced increase in Golgi-localized Optn. *Atg5*[KO] MEFs were treated

with etoposide (10 μM) for 4 or 12 hr, and cells were immunostained with anti-Optn (green) and anti-Golgin97 (red) antibodies. Representative images are shown in (**e**) Bars = 10 μm. ROIs are indicated as squares, and their magnified images are shown in the right panels. In **f** colocalization of the fluorescence signals of the two molecules (Optn and Golgin97) at different time points after etoposide treatment was evaluated using Pearson correlation coefficient. Data are shown as the mean ± SE (*n* = 10 cells, experiments performed two times). **g, h** Similar experiments to (**e, f**) were performed using an anti-Lamp1 antibody (red) instead of an anti-Golgin97 antibody. Etoposide-induced an increase in Golgi-localized Optn. Data are shown as the mean ± SE (*n* = 10 cells, experiments performed two times). In **b, d, f** and **h** comparisons were performed using one-way ANOVA followed by the Tukey's *post-hoc* test. *$p < 0.05$, **$p < 0.01$; ns: not significant. Data are representative of three independent experiments in (**a, c**) and two independent experiments in (**e, g**).

added to the substrate molecule, which becomes an eat-me signal, and the ubiquitinated proteins are incorporated into autophagosomes[14,15]. Ubiquitination is one of the most common post-translational modifications in a cell, and Ub chains can be linked via any of its seven Lys residues (K6, K11, K27, K29, K33, K48, and K63) or via the amino terminal M1 residue (generating a linear chain)[16–18]. It is well known that polyUb bound to K48 acts as a signal for protein degradation by the proteasome, and that polyUb bound to K63 acts as a signal for protein degradation by autophagy[19–22]. In addition to the major Ub chains, such as K48, K63, and M1 that have been analyzed to date, progress has been made recently in the analysis of atypical Ub chains, which are involved in specific intracellular processes and signal transduction pathways with diverse functions. For example, K6-linked chains are essential for DNA repair[23], whereas K11-linked chains regulate the cell cycle, particularly during the progression and termination of mitosis[24], and control protein degradation during cell division. K27- and K29-linked chains are involved in immune responses and inflammatory signaling, with K29-linked chains specifically aiding in the removal of abnormal proteins, such as Mitochondrial antiviral signaling, to support immune system function[25,26]. Additionally, K33-linked chains contribute to intracellular transport and signal transduction regulation by ensuring the proper localization of specific proteins[27]. Together, these atypical Ub chains are crucial for maintaining cellular homeostasis and facilitating adaptive responses.

In selective autophagy, ubiquitinated molecules interact with Ub-recognizing adaptor proteins, including p62/SQSTM1, optineurin (OPTN), TAX1BP1, NDP52, and NBR1, and are incorporated into autophagosomes to be degraded[28–32]. These five molecules cover almost all substrates, and hence autophagic degradation is severely impaired in cells lacking all five of these molecules[33,34]. Unlike autophagy, the mechanism underlying the recognition of substrates in GOMED remains unidentified to date. Therefore, we here investigated whether GOMED degrades substrates selectively, and also investigated its molecular mechanism. We found that selective protein degradation occurs in GOMED by the recognition of substrates using K33-Ub chains as an eat-me signal, and OPTN as an adaptor molecule. Our results led to the identification of a third mechanism of Ub-mediated proteolysis, namely, K33-Ub chain-mediated GOMED, in addition to the already known K48-Ub chain-mediated proteasome degradation and linear Ub/K63-Ub chain-mediated autophagy[35–37]. In vivo, we also found that OPTN-mediates GOMED functions in mitochondrial removal during the terminal differentiation of erythrocytes.

## Results

### OPTN is required for GOMED induced by genotoxic stress

GOMED is a proteolysis mechanism that mainly degrades proteins that have passed through the Golgi, by generating autophagosome-like and autolysosome-like structures[4]. Because the mechanism by which GOMED selectively traps substrate proteins to deliver them to autophagosome-like structures has not yet been elucidated, we here analyzed whether substrate-recognition adaptor proteins are utilized

in GOMED, similarly to in selective autophagy. We first investigated the possible involvement of the autophagy adaptor proteins p62, NDP52, NBR1, TAX1BP1, and OPTN in GOMED, by analyzing their expression levels after GOMED induction. For this experiment, we used *Atg5*-deficient (*Atg5*[KO]) mouse embryonic fibroblasts (MEFs) to avoid any effects of autophagy and to easily detect GOMED[4], and added etoposide, which is a GOMED inducer via genotoxic stress. Among the five autophagy adaptor proteins, only Optn was transiently increased and then decreased in response to etoposide treatment, and this reduction was abolished by the concomitant addition of chloroquine (CQ), which is an inhibitor of lysosomal degradation (Fig. 1a, b), suggesting that Optn is a GOMED substrate and a candidate adaptor protein for the recognition of GOMED substrates. To investigate whether Optn is a simple substrate or functions as an adaptor protein in GOMED, we silenced *Optn* and analyzed the extent of GOMED using DAPGreen, which is a newly developed probe[38] that detects both autophagy and GOMED[39] (Supplementary Fig. 1). As expected, DAPGreen signals increased upon etoposide treatment in *Atg5*[KO] MEFs owing to the induction of GOMED, and this increase was substantially reduced by *Optn* silencing (Fig. 1c, d, and Supplementary Fig. 2a), supporting the possibility that Optn functions as an adaptor protein. To strengthen this observation, we performed rescue experiments using four different siRNAs targeting Optn (siRNAs no.1–4) and their respective resistant gene created by substituting bases without altering the amino acid sequence (Flag-Optn R1–R4). As indicated, each siRNA achieved successful gene knockdown, and Optn expression (particularly 2 to 4) was well recovered by their respective resistant gene (Supplementary Fig. 2b). In these cells, the number of DAPGreen puncta induced by etoposide was substantially reduced by Optn siRNA, and recovered by their respective resistant plasmid (Supplementary Fig. 2c, d), confirming the important role of Optn as an adaptor protein in etoposide-induced GOMED. Therefore, the loss of Optn is expected to decrease the extent of GOMED by reducing the amount of recognized substrates.

As GOMED originates from *trans*-Golgi membranes, GOMED adaptor proteins are expected to colocalize with *trans*-Golgi membranes[4], and we indeed found an increase in the colocalization of Optn with Golgin97, which is a protein localized on *trans*-Golgi membranes, upon etoposide treatment (Fig. 1e, f). GOMED structures eventually colocalized with lysosomes to become autolysosome-like structures, and we also observed the colocalization of Optn with Lamp1, a marker of lysosomes, after the addition of etoposide (Fig. 1g, h).

The crucial role of OPTN was also analyzed using HeLa cells and Pentaknockout (KO) HeLa cells lacking p62, NDP52, NBR1, Tax1BP1, and OPTN[33,34]. For this purpose, we first expressed tagRFP-LC3 to these cells, treated them with etoposide, and then analyzed the formation of LC3 and DAPGreen puncta in the same cell. DAPGreen can recognize almost all autophagic structures from the phagophore to the autolysosome, as well as almost all GOMED structures (Supplementary Fig. 1d). Therefore, LC3-positive DAPGreen puncta are considered autophagic structures, while LC3-negative DAPGreen puncta are

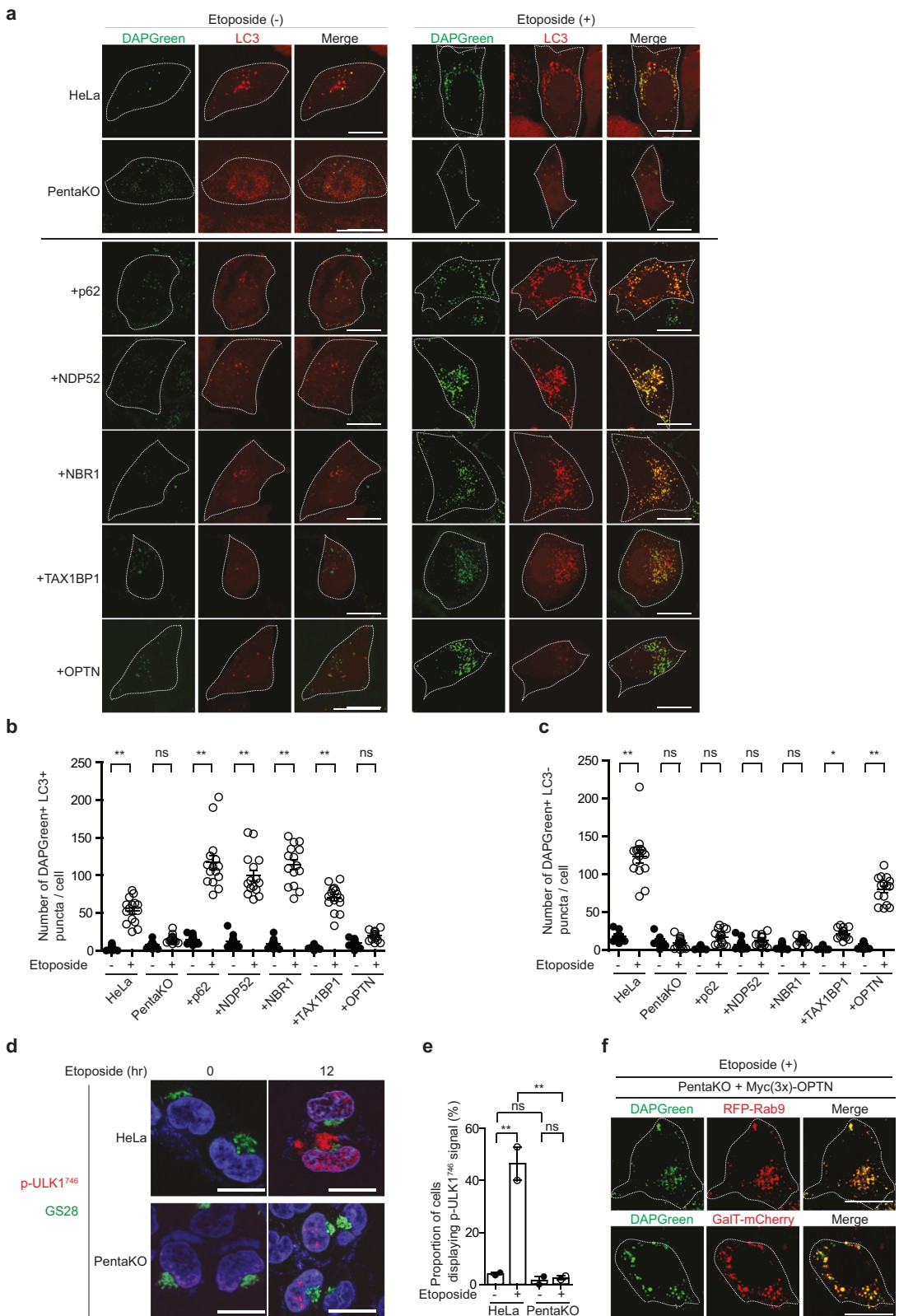

recognized as structures associated with GOMED. This has already been confirmed using the CLEM (Correlative Light and Electron Microscopy) assay[38,39] (Supplementary Fig. 1). Additionally, a small number of LC3-positive but DAPGreen-negative puncta may appear, which are thought to represent non-autophagic LC3 puncta, such as those associated with LC3-associated phagocytosis (LAP). As a result of experiment, we found that both the number of DAPGreen$^+$/LC3$^+$

puncta (autophagic puncta) and the number of DAPGreen$^+$/LC3$^-$ puncta (GOMED puncta) were increased in HeLa cells, but not in PentaKO cells (Fig. 2a–c). Less GOMED occurring in PentaKO cells than control HeLa cells was confirmed by the absence of Ulk1$^{746}$ phosphorylation signals, which is a specific marker of GOMED[7] (Fig. 2d, e), suggesting that at least one of the autophagy adaptor proteins is involved in GOMED.

**Fig. 2 | OPTN is required for GOMED but not for canonical autophagy in etoposide-treated HeLa cells. a**–**c** HeLa cells and PentaKO cells were transfected with plasmids encoding LC3-TagRFP, together with or without Myc (3×)-tagged-adaptor proteins (p62/SQSTM1, NDP52, NBR1, TAX1BP1, or OPTN). After 24 hr, cells were treated with or without etoposide (10 μM) for 12 hr, and analyzed using DAPGreen. Representative images of LC3-TagRFP and DAPGreen are shown in (**a**). White dotted lines indicate the cell shapes. Scale bars = 10 μm. Quantitative analysis of DAPGreen$^+$/LC3$^+$ puncta (**b**) and DAPGreen$^+$/LC3$^-$ puncta (**c**) are shown. Data are shown as the mean ± SE **b** $n = 15$, **c** $n = 15$. **d**, **e** HeLa cells and PentaKO cells were treated with or without etoposide (10 μM) for 12 hr, and immunostained with anti-p-ULK1$^{746}$ (red) and anti-GS28 (green) antibodies. Nuclei were counterstained with DAPI. In **d** representative images are shown. Bars = 10 μm. p-ULK1$^{746}$ signals were observed on the Golgi upon GOMED induction. In **e** the proportion of cells with p-ULK1$^{746}$ signals was calculated. Data are shown as the mean ± SE ($n \geq 10$ cells). Note that the anti-p-ULK1$^{746}$ antibody is produced against phosphoserine$^{746}$ of mouse ULK1, but it also recognizes phosphoserine$^{747}$ of human ULK1. In **f** PentaKO cells were transfected with plasmids encoding Myc (3×)-tagged-OPTN together with RFP-Rab9 or GalT-mCherry. After 24 hr, cells were treated with etoposide (10 μM) for 12 hr and analyzed using DAPGreen. Representative images are shown. Scale bars = 10 μm. In **b**, **c**, **e** comparisons were performed using one-way ANOVA followed by the Tukey's *post-hoc* test. **$p < 0.01$; ns: not significant. Data are representative of three independent experiments in (**a**) and two independent experiments in (**d**, **f**).

To identify molecule(s) involved in GOMED, we transfected plasmids encoding each of the adaptor proteins into PentaKO cells, and treated them with etoposide. Each protein was successfully expressed in PentaKO cells, with an expression level nearly the same as or less than the endogenous level in control HeLa cells (Supplementary Fig. 3a, b). Interestingly, the expression of each of p62, NDP52, NBR1, and TAX1BP1 restored the formation of DAPGreen$^+$/LC3$^+$ puncta, but not DAPGreen$^+$/LC3$^-$ puncta (Fig. 2a–c), indicating that all the adaptor proteins excluding OPTN play a role in etoposide-induced autophagy. In contrast, the expression of OPTN restored only DAPGreen$^+$/LC3$^-$ puncta (Fig. 2a, c), indicating that OPTN is involved in etoposide-induced GOMED in HeLa cells. The OPTN-dependent DAPGreen puncta were well merged with Rab9 (a crucial protein for GOMED) and GalT-mCherry (a Golgi marker) in etoposide-treated cells (Fig. 2f, Supplementary Fig. 4a–d), supporting that the DAPGreen puncta are GOMED structures. Some of the DAPGreen also colocalized with lysosomes, which is thought to represent autolysosome-like GOMED structures (Supplementary Fig. 4e, f). Regarding etoposide-induced autophagy, OPTN does not appear to be involved because LC3 puncta were not observed even after the blockage of autophagy flux by E64d and pepstatin treatment, inhibitors of lysosomal protein degradation (Supplementary Fig. 5). All these data indicated that OPTN is involved in etoposide-induced GOMED, but not in autophagy in HeLa cells. The involvement of OPTN in GOMED is reasonable, because OPTN is known to localize to the Golgi[40,41], and GOMED membranes originate from Golgi membranes[4,6].

## Involvement of OPTN in 1, 3-cyclohexanebis(methylamine) (CBM)-induced GOMED

We next aimed to clarify whether OPTN is involved in GOMED induced by another stimulus. For this purpose, we treated *Atg5*$^{KO}$ MEFs with CBM, which inhibits protein trafficking from the Golgi to the plasma membrane, and thereby induces GOMED but not autophagy[6]. We observed the CBM-induced transient increase and subsequent decrease of Optn and its restoration by CQ (Fig. 3a, b), as observed with etoposide treatment (Fig. 1a). Furthermore, CBM increased the number of DAPGreen puncta in *Atg5*$^{KO}$ MEFs, which was substantially suppressed by *Optn* silencing (Fig. 3c, d), indicating that Optn is involved in CBM-induced GOMED. We also treated HeLa cells with CBM, and found that autophagy was not induced as assessed by LC3 puncta formation (Fig. 3e, f). GOMED was analyzed using LAMP2 immunostaining instead of DAPGreen, because LAMP2 signals appear as small puncta in lysosomes, but when they fuse with autophagosome-like GOMED structures (or autophagosomes) to form autolysosome-like GOMED structures (or autolysosomes), they can be observed as larger puncta (which are sometimes ring-shaped). The correspondence of large LAMP2 puncta to autolysosome-like GOMED structures (or autolysosomes) was previously confirmed using CLEM analysis[6,42]. Therefore, this characteristic can be used as an indicator to evaluate the abundance of autolysosome-like GOMED structures (or autolysosomes). We found that many large LAMP2 puncta were generated upon CBM treatment in HeLa cells, whereas only a few were generated in PentaKO cells (Fig. 3e, g). Consistent results were obtained by LAMP2 immunoblot analysis (Supplementary Fig. 6). Furthermore, the introduction of OPTN, but not other adaptor proteins, increased the number of large LAMP2 puncta in CBM-treated PentaKO cells (Fig. 3h, i), suggesting that OPTN functions as an adaptor protein for CBM-induced GOMED, as in etoposide-induced GOMED.

## Role of OPTN in the recognition of ubiquitinated substrates

OPTN is known to interact with Ub and to recruit ubiquitinated substrates to autophagosomes for degradation by autophagy[29,43], and we hence hypothesized that it has the same role in GOMED. We previously reported that vesicular stomatitis virus G protein VSV-Gt045 (VSVG)-GFP[6], a widely used artificial protein for analyzing Golgi trafficking, is degraded by GOMED upon etoposide and CBM treatment. Therefore, we aimed to clarify whether VSVG-GFP is ubiquitinated and thereby interacts with OPTN during GOMED. For this purpose, we used *Atg5*$^{KO}$ MEFs stably expressing VSVG-GFP, treated them with CBM, and performed immunoprecipitation of VSVG-GFP using an anti-GFP antibody. The immunoprecipitants were then immunoblotted against total Ub (PanUb), and we observed the CBM-induced ubiquitination of VSVG-GFP in *Atg5*$^{KO}$ MEFs (Fig. 4a, lanes 2, 3). Ubiquitination was validated by the complete loss of PanUb signals using the specific E1 Ub activating enzyme inhibitor TAK-243[44] (Fig. 4a, lanes 6, 7), and by the presence of Ub signals even upon denaturing immunoprecipitation (Supplementary Fig. 7a, lane 8). Furthermore, when we analyzed whether Optn was coimmunoprecipitated with VSVG-GFP in CBM-treated *Atg5*$^{KO}$ MEFs, as expected, Optn, but none of the other four adaptor proteins, was eluted as a VSVG-GFP-interacting protein (Fig. 4b, lane 5, Fig. 4c). The interaction with Optn was substantially inhibited by the addition of an E1 inhibitor (Fig. 4b, lane 6, Fig. 4c), and was not observed in untreated cells (Fig. 4b, lane 4, Fig. 4c), indicating that Optn interacts with ubiquitinated VSVG-GFP upon CBM treatment. Note that it has been reported that the VSVG portion of VSVG-GFP is facing the cytosolic side on the vesicle of the Golgi apparatus, the cytosolic side of VSVG contains several Lys residues[45,46], and it is suggested that this region is likely to undergo ubiquitination[45,46]. Consistently, when only VSVG was expressed, VSVG was ubiquitinated (Supplementary Fig. 1b, lanes 10, 11) and OPTN was coimmunoprecipitated with VSVG (Supplementary Fig. 7b, lanes 4, 5), and these activities were abolished by TAK-243 (Supplementary Fig. 7b, lanes 6, 12). In contrast, GFP on its own undergoes only minimal ubiquitination compared with VSVG-GFP, and no binding to OPTN was observed (Supplementary Fig. 7c, lanes 5, 6).

Consistently, immunofluorescence analysis revealed that a proportion of VSVG-GFP was accumulated in the Golgi upon CBM treatment (owing to the blockage of trafficking from the Golgi to the plasma membrane) and was colocalized with Optn (Fig. 4d, e). Furthermore, a proportion of VSVG-GFP was also colocalized with Lamp1 to be degraded in autolysosome-like GOMED structures in the cells expressing Optn (Fig. 4f, g). Note that this colocalization was not observed in the absence of Optn, and was increased by the addition of CQ (Fig. 4f, g). These data suggested that VSVG-GFP interacts with Optn in the Golgi, by which VSVG-GFP is delivered to autolysosome-like GOMED

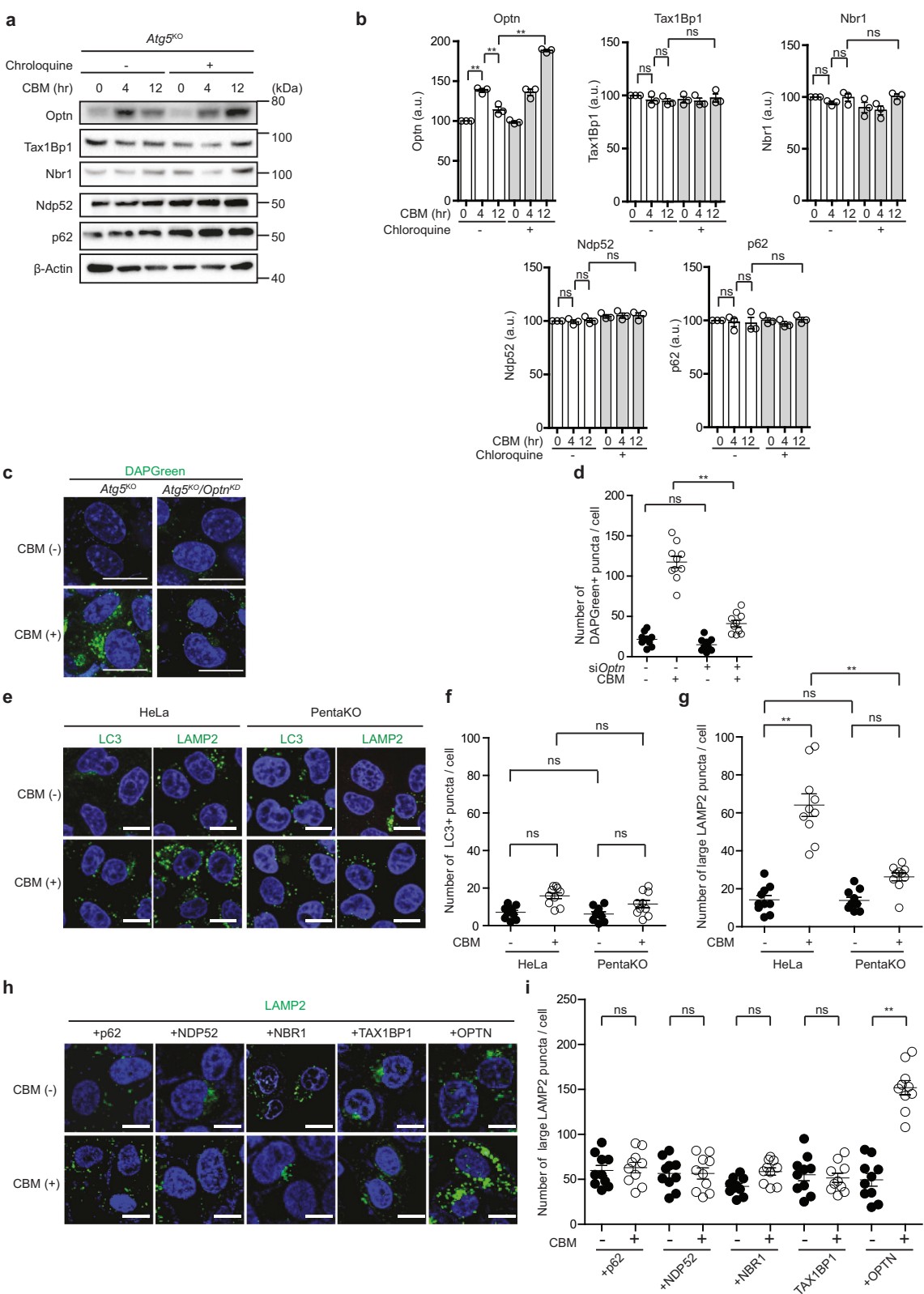

structures. Biochemical analysis confirmed this scenario. During GOMED, VSVG-GFP is first cleaved at the junction of the two proteins and subsequently the GFP part is degraded, so that a transient GFP cleavage band appears. Indeed, the cleaved GFP band increased time-dependently after CBM treatment, and was decreased by TAK-243 treatment (Fig. 4a–c). This cleavage was also suppressed by the addition of CQ (Supplementary Fig. 8). Importantly, these cleaved bands were substantially reduced by the silencing of *Optn* (Fig. 4h, i). Taken together, Optn accumulates in the Golgi apparatus, and functions as an adaptor protein for ubiquitinated VSVG-GFP in GOMED (Fig. 4j).

**K33 polyubiquitin is crucial for VSVG-GFP recognition by OPTN**
As there are various types of Ub chains, we next aimed to determine the type of Ub chain(s) required for VSVG-GFP recognition by OPTN.

**Fig. 3 | Involvement of OPTN in CBM-induced GOMED. a** The dynamics of autophagy adaptor proteins after CBM treatment. $Atg5^{KO}$ MEFs were treated with CBM (1 mM) in the presence or absence of Chroloquin (100 µM). At the indicated times, cells were lysed, and the expression of each protein was analyzed by western blotting. β-actin was used as a loading control. **b** Semiquantitative analysis of protein expression in (**a**) is shown. Data are shown as the mean ± SE ($n = 3$). **c, d** Involvement of Optn in CBM-induced GOMED in $Atg5^{KO}$ MEFs. *Optn* was silenced in $Atg5^{KO}$ MEFs, and the indicated MEFs were treated with CBM (1 mM) for 12 hr. Then, the extent of GOMED was analyzed using DAPGreen. Nuclei were counterstained with Hoechst 33342. Representative images are shown in (**c**). Bars = 10 µm. In **d** the number of DAPRed puncta per cell was calculated. Data are shown as the mean ± SE ($n = 10$ cells, experiments performed 2 times). **e–g** HeLa cells and PentaKO cells were treated with CBM (1 mM) for 12 hr or left untreated, and then immunostained with anti-LC3 and anti-LAMP2 antibodies. Nuclei were counterstained with DAPI. Representative images are shown in (**e**). Scale bars = 10 µm.

Quantitative analyses of LC3 puncta (**f**) and LAMP2 puncta (**g**) were performed. Data are shown as the mean ± SE ($n = 10$ cells, experiments performed 2 times). **h, i** Involvement of OPTN in CBM-induced GOMED in HeLa cells. The indicated proteins were transiently expressed in PentaKO cells. After 24 hr, cells were treated with CBM (1 mM) for 12 hr, or left untreated. Then, the extent of GOMED was analyzed using an anti-LAMP2 antibody (green). Nuclei were counterstained with DAPI. Representative images are shown in **h**. Bars = 10 µm. In **i** the number of large LAMP2 puncta per cell was calculated. Data are shown as the mean ± SE ($n = 10$ cells, experiments were performed two times). In (**b**), comparisons were performed using the unpaired two-tailed Student $t$-test and one-way ANOVA followed by the Tukey's *post-hoc* test. In (**d, f, g, i**) comparisons were performed using one-way ANOVA followed by the Tukey's *post-hoc* test. *$p < 0.05$, **$p < 0.01$; ns: not significant. Data are representative of three independent experiments in (**a**) and two independent experiments in (**c, e,** and **h**).

To this end, we expressed various types of Myc (3×)-tagged Ub chains and VSVG-GFP into HEK293T cells, treated them with CBM for 12 hr, and then analyzed the Ub chains interacting with VSVG-GFP. Ub has seven Lysine residues, and the Ub in which all these Lys residues are mutated to Arginine (Arg) is defined as K0. Additionally, Ub in which only the x-th Lys residue remains is referred to as Kx. Furthermore, Ub in which only the x-th Lys is mutated to Arg is referred to as KxR. Among the various Ub chains, a large amount of K29-polyUb, K33-polyUb and K63-polyUb were detected in cell lysates (Fig. 5a, lanes 5, 6, and 8). Among them, K33-polyUb was substantially accumulated in the immunoprecipitant of VSVG-GFP (Fig. 5a, lane 15), indicating that K33-polyUb is conjugated to VSVG-GFP during CBM-induced GOMED. Weaker signals of K33R supported the involvement of K33-Ub in CBM-induced GOMED (Fig. 5b, lanes 14, 15). We also observed the colocalization of VSVG-GFP and K33-Ub, but not K33R-Ub, upon the treatment of $Atg5^{KO}$ MEFs with CBM (Fig. 5c), indicating that CBM induced the K33-polyubiquitination of VSVG-GFP. These results suggest that K33-polyubiquitinated VSVG-GFP is trapped by OPTN upon CBM treatment. In fact, when we immunoprecipitated Flag-OPTN, K33-polyUb was detected (Fig. 5d, lane 10), and the signal was increased upon CBM treatment (Fig. 5d, lane 14). Similar to CBM treatment, when we expressed various types of Myc (3×)-tagged Ub chains together with Flag-OPTN into HEK293T cells, treated them with etoposide for 12 hr, and immunoprecipitated the lysates with Flag-OPTN, we found that K33-Ub strongly and K63-Ub weakly interacts with Flag-OPTN in these etoposide-treated cells (Supplementary Fig. 9a, lanes 15 and 17). Because the K63R signal appeared to be slightly weak (Supplementary Fig. 9b, lane 15), this suggests that in addition to K33-Ub, K63-Ub functions in GOMED upon etoposide stimulation. However, as etoposide stimulation induces both autophagy and GOMED, it is possible that K63 is only involved in autophagy. In fact, upon CBM stimulation, which induces only GOMED, there is no involvement of K63-Ub (the polyUb signal seen in K63R should indicate K33-Ub) (Supplementary Fig. 9c). Therefore, the involvement of K63-polyUb in GOMED is minimal, and K33 Ub chains are crucial for substrate recognition by OPTN during GOMED.

To confirm the involvement of K33-polyUb, we analyzed the effect of the silencing of *Trabid* (Zranb1), a K29/K33 chain-specific deubiquitinase (DUB)[47,48]. *Trabid* knockdown was confirmed by western blotting (Supplementary Fig. 10), and this was expected to upregulate GOMED activity because of the increase in K33-polyubiquitinated substrate proteins. In fact, when we analyzed the extent of GOMED using DAPRed, a newly developed red-fluorescent probe for the detection of autophagy and GOMED (Supplementary Fig. 1b, d), the number of DAPRed puncta was substantially increased by *Trabid* silencing regardless of GOMED stimuli (Fig. 5e, f). VSVG-GFP degradation was also increased (Supplementary Fig. 10, lane 4), supporting the idea that K33-polyUb is involved in GOMED. These data indicated

that OPTN recognizes K33-polyubiquitinated substrate proteins for their degradation by GOMED.

## Endogenous GOMED substrates integrin α5 and mesothelin are also degraded via the K33-polyUb-OPTN pathway

Next, we investigated whether the same substrate recognition mechanism also functions for endogenous substrates. Integrin α5, a molecule that is transported to focal adhesions via the Golgi apparatus, was previously reported by us to be degraded through GOMED[7]. Therefore, we analyzed this molecule as an endogenous GOMED substrate. We first investigated whether integrin α5 is also degraded via autophagy. Treatment of HeLa cells with rapamycin, which induces only autophagy and not GOMED, did not lead to a substantial change in the fluorescence signal of integrin α5 (calculated as fluorescence intensity × area per cell) (Supplementary Fig. 11a, b). Although rapamycin increased the number of LC3 puncta (Supplementary Fig. 11a, c), colocalization between LC3 puncta and integrin α5 was rarely observed (Supplementary Fig. 11a, d). These findings suggest that integrin α5 is not a substrate of autophagic degradation. In contrast, treatment of HeLa cells with CBM, which selectively blocks post-Golgi transport and induces GOMED, led to an increase in the intracellular fluorescence signal of integrin α5. Cotreatment with E64d/pepstatin further enhanced this signal (Supplementary Fig. 12a, b). Additionally, colocalization of integrin α5 with LAMP1 increased upon CBM treatment, and was further enhanced by E64d/pepstatin treatment (Supplementary Fig. 12a, c), whereas colocalization with LC3 did not occur even after E64d/pepstatin treatment (Supplementary Fig. 12a, d). These results indicate that integrin α5 is degraded by GOMED in cells treated with CBM. This is further supported by the finding that integrin α5 signals colocalizing with LAMP1 were not observed in PentaKO cells, which have low GOMED activity (Supplementary Fig. 12a, c).

We next investigated whether integrin α5 is degraded by GOMED in cells treated with etoposide, which induces both autophagy and GOMED. In addition to wild-type HeLa cells, we used PentaKO cells, and HexaKO cells, which are deficient for all LC3 family members and thus incapable of undergoing autophagy. In wild-type HeLa cells, etoposide treatment, owing to its inhibitory effect on Golgi-mediated transport, slightly increased the intracellular levels of integrin α5, and the further addition of E64d/pepstatin increased integrin α5 levels even more (Fig. 6a, b). Similar increases were also observed in HexaKO cells. In contrast, integrin α5 levels in PentaKO cells were high even at the basal level, and did not increase any further upon etoposide and E64d/pepstatin treatment (Fig. 6a, b). These results indicate that integrin α5 is primarily degraded by GOMED rather than autophagy. Moreover, we previously demonstrated that GSK872, a RIP3 inhibitor, suppresses GOMED without affecting autophagy[7]. Based on this finding, we inhibited GOMED in HeLa cells using GSK872, which resulted in increased intracellular levels of integrin α5, similar to what was observed in PentaKO cells. Furthermore, no additional increase in

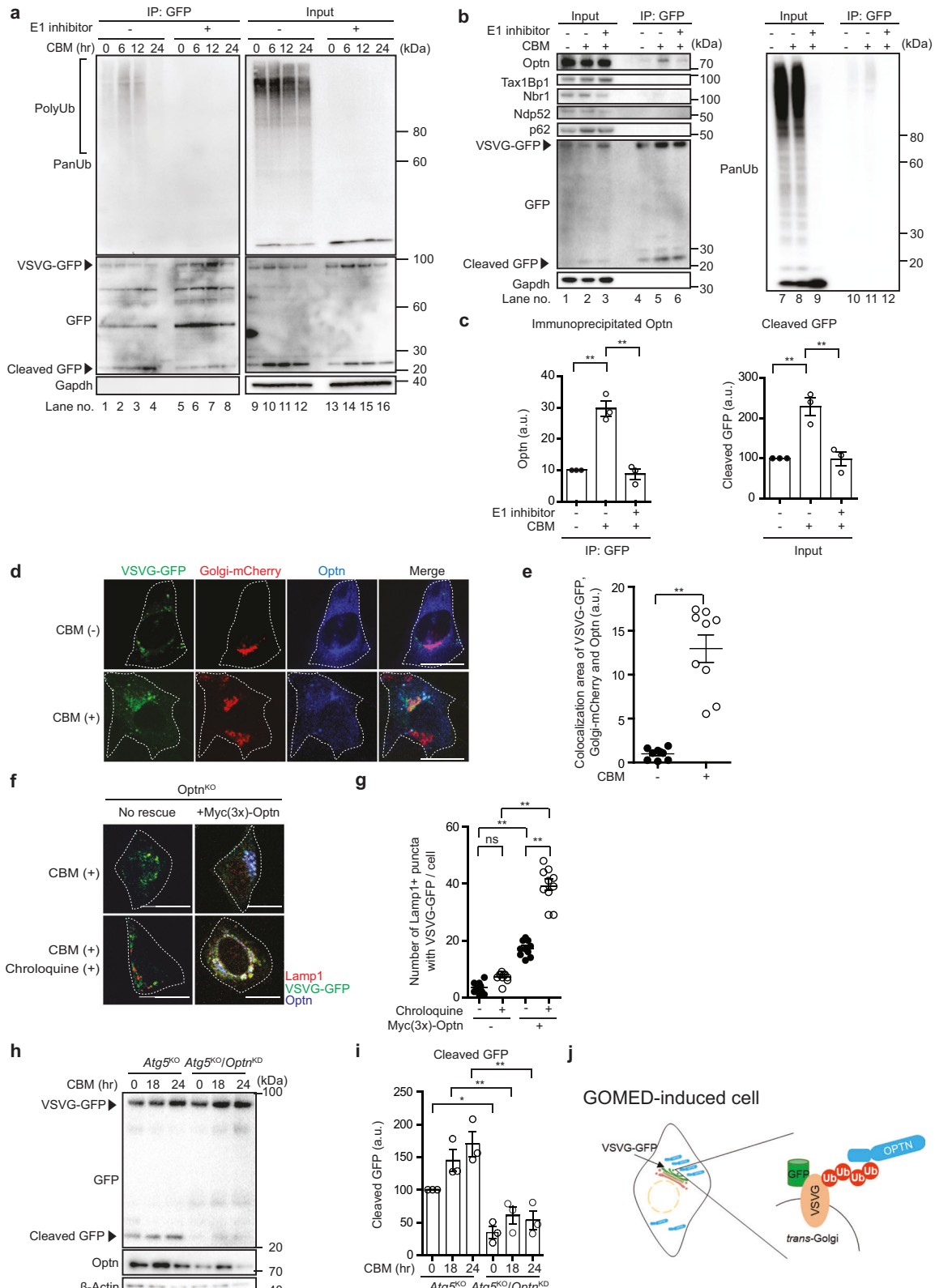

integrin α5 levels was observed following E64d/pepstatin treatment (Fig. 6a, b). This also supports that integrin α5 degradation occurs via GOMED.

These findings were further confirmed by immunostaining. If integrin α5 is degraded via GOMED, we would expect it to colocalize with Lamp2 but not with LC3. Indeed, etoposide treatment of wild-type HeLa cells resulted in the appearance of approximately 120 integrin

α5–LAMP2 colocalizing puncta per cell, which increased to approximately 180 puncta after E64d/pepstatin treatment (Fig. 6c, d). In contrast, very few LC3-positive puncta colocalizing with integrin α5 were observed (Fig. 6c, e). The few integrin α5–LC3 colocalizing puncta that were detected likely represent fusion products of autophagy-derived LC3-positive autolysosomes and autolysosome-like GOMED structures. Supporting this, knockdown of syntaxin 17

**Fig. 4 | VSVG-GFP, a GOMED substrate, is ubiquitinated and then interacts with Optn. a** Polyubiquitination of VSVG-GFP upon CBM treatment. VSVG-GFP-expressing *Atg5*KO MEFs were treated with CBM (1 mM) for the indicated times. Cells were also treated with or without the E1 inhibitor TAK-243 (100 nM) for 1 hr before cell collection. Cells were then lysed and immunoprecipitated with an anti-GFP antibody. Expression of polyUb, VSVG-GFP/GFP, and Gapdh in immunoprecipitants (IP: GFP) and total lysates (Input) were analyzed by western blotting. Note that VSVG-GFP was ubiquitinated and cleaved upon CBM treatment. **b** Interaction of ubiquitinated VSVG-GFP with Optn. VSVG-GFP-expressing *Atg5*KO MEFs were treated with CBM (1 mM) for 12 hr. Cells were also treated with or without TAK-243 (100 nM) for 1 hr before cell collection. Cells were then lysed and immunoprecipitated with an anti-GFP antibody. Expression of the autophagy adaptor proteins, VSVG-GFP/GFP, Gapdh, and polyUb in total lysates (Input) and immunoprecipitants (IP: GFP) were analyzed by western blotting. Note that only Optn was coimmunoprecipitated with VSVG-GFP upon CBM treatment. **c** Semiquantitative analysis of immunoprecipitated Optn (left panel) and cleaved GFP from VSVG-GFP (right panel) in (**b**) is shown. Data are shown as the mean ± SE (*n* = 3). **d, e** Colocalization of VSVG-GFP and Optn on the Golgi upon CBM treatment. Golgi-mCherry-expressing *Atg5*KO MEFs transiently expressing VSVG-GFP, were treated with CBM (1 mM) for 12 hr or left untreated. Cells were then immunostained with an anti-Optn antibody, and images of VSVG-GFP (green), Golgi (red), and Optn (blue) were obtained. Representative images are shown in **d**. Bars = 10 μm. White dotted lines

indicate the cell shapes. In **e** the area of colocalization between VSVG-GFP and Optn on Golgi-mCherry at different time points after CBM treatment was calculated using ImageJ software. Data are shown as the mean ± SE (*n* = 10 cells, experiments were performed two times). **f, g** In *Optn*KO MEF cells, Myc(3×)-Optn and VSVG-GFP were transiently expressed. After 24 hr, cells were treated with CBM (1 mM) together with or without chroloquin (100 μM) for 24 hr, and were immunostained with antibodies against Lamp1 (red), GFP (green) and Myc (blue). Colocalization of VSVG-GFP and Lamp1 was observed in the cells expressing Optn (blue). In **g**, the number of Lamp1 puncta colocalizing with VSVG-GFP per cell was calculated. Data are shown as the mean ± SE (*n* = 10 cells). **h.** Involvement of Optn in CBM-induced VSVG-GFP cleavage in *Atg5*KO MEFs. *Optn* was silenced in VSVG-GFP-expressing *Atg5*KO MEFs, and the indicated MEFs were treated with CBM (1 mM) for the indicated times. Then, cells were lysed and the extent of VSVG-GFP cleavage and the expression of Optn was analyzed by western blotting. β-actin was used as a loading control. **i.** Semi-quantitative analysis of protein expression in **h**. Data are shown as the mean ± SE (*n* = 3). **j.** Schematic diagram showing that OPTN accumulates at the Golgi apparatus, where it functions as an adaptor protein for ubiquitinated VSVG-GFP in GOMED. This scheme was designed and created by the authors using Microsoft PowerPoint 365. In **c, d, e, g,** and **i**, comparisons were performed by one-way ANOVA followed by the Tukey's *post-hoc* test. **p < 0.01; ns: not significant. Data are representative of three independent experiments in (**a, b and h**) and two independent experiments in (**d and f**).

(STX17), a key factor in autolysosome formation[39], completely abolished integrin α5–LC3 colocalizing puncta (Supplementary Fig. 13a, b). Furthermore, when GOMED was inhibited by GSK872, integrin α5–LAMP2 colocalizing puncta were rarely observed (Fig. 6c, d), and the intracellular fluorescence intensity of integrin α5 increased (Fig. 6c, f), which is consistent with the results of western blot analysis (Fig. 6a, b). In addition, integrin α5–LC3 colocalizing puncta were also rarely observed (Fig. 6c, e). These data strongly suggest that integrin α5 is not degraded via autophagy, but is instead degraded via GOMED upon etoposide treatment. Furthermore, the large amount of colocalization of integrin α5 with Rab9 supports the idea that integrin α5 is degraded by GOMED (Supplementary Fig. 14a).

We next investigated whether the endogenous GOMED substrate integrin α5 is polyubiquitinated via K33-linked ubiquitin chains and then recognized by OPTN. In *Atg5*KO MEFs treated with etoposide, integrin α5 colocalized with OPTN on the Golgi apparatus in an etoposide-dependent manner (Fig. 6g, h), and also colocalized with lysosomes (Supplementary Fig. 14b, c). Knockdown of *Optn* resulted in an increase in integrin α5 levels (Fig. 6i). Although integrin α5 showed weak constitutive binding with OPTN in cells under basal conditions (Fig. 6j, lane 5), this was enhanced upon etoposide treatment (Fig. 6j, lanes 6, 7), and was markedly reduced by the treatment of cells with TAK-243 (Fig. 6j, lane 8). Additionally, K33-linked ubiquitin interacted with integrin α5, whereas the K33R mutant did not (Fig. 6k, lanes 14–15). These results indicate that OPTN and K33-linked polyubiquitin are involved in the GOMED-mediated degradation pathway of integrin α5.

We further analyzed another endogenous GOMED substrate, mesothelin (MSLN), which is transported to the plasma membrane via the Golgi apparatus. Similar to integrin α5, MSLN expression increased upon E64d/pepstatin treatment in both HeLa and HexaKO cells, whereas in PentaKO cells, its expression was high even under basal conditions (Supplementary Fig. 15a, b). In etoposide-treated HeLa cells, MSLN did not colocalize with LC3, but showed clear colocalization with LAMP2 (Supplementary Fig. 15c–f), suggesting that MSLN is a degradation substrate of GOMED. MSLN also colocalized with OPTN and Rab9 in an etoposide-dependent manner, and the simultaneous addition of bafilomycin A1 increased their colocalization (Supplementary Fig. 15g–j). Etoposide stimulation induced colocalization of MSLN with LAMP2 and OPTN, but to a lesser extent than with substrates such as integrinα5 and VSVG, possibly due to suboptimal stimulation conditions. Additionally, knockdown of *Optn* increased Msln

levels (Supplementary Fig. 15k); the interaction between Msln and Optn was etoposide-dependent and reduced by TAK-243 treatment (Supplementary Fig. 15l); and MSLN interacted with K33-linked ubiquitin chains (Supplementary Fig. 15). These findings, consistent with those for integrin α5, indicate that MSLN is recognized by OPTN and degraded via the GOMED pathway in a K33-ubiquitin-dependent manner.

## OPTN recognizes polyubiquitin via its zinc-finger domain in GOMED

We further searched for the region of OPTN that is crucial for its recognition of Ub[49,50]. Because OPTN contains two Ub-interacting domains, namely, the Ub-binding domain (UBD) and the zinc-finger (ZF) domain (Fig. 7a)[51], we generated three mutant OPTNs lacking either the UBD, ZF, or both (Q398X). To analyze their activity in GOMED, we expressed Flag-OPTN or its mutants in PentaKO cells, treated the cells with etoposide, and assessed GOMED by evaluating large LAMP2 puncta formation. As functional adaptor molecules are required for GOMED induction, large LAMP2 fluorescence puncta were observed in cells expressing wild-type OPTN, but not vector-transfected PentaKO cells (Fig. 7b, c), consistent with Fig. 2. Importantly, the UBD-deletion mutant showed similar activity to wild-type OPTN, and the other mutants had no activity (Fig. 7b, c), indicating that the ZF domain is crucial for GOMED induction. Consistently, when HEK293T cells were treated with etoposide, and the amount of Ub within Flag-OPTN immunoprecipitants was analyzed, we found strong Ub signals in the cells expressing wild-type OPTN and the UBD-deletion mutant, but not mutants lacking the ZF domain (Fig. 7d), indicating the crucial role of the ZF domain in the recognition of Ub chains. Taken together, OPTN is important for the recognition of K33-polyubiquitinated substrates via its ZF domain in GOMED (Fig. 7e).

## OPTN is involved in mitochondrial elimination during erythroid maturation

To clarify the effect of OPTN in vivo, we analyzed erythrocytes, because GOMED degrades the mitochondria of erythrocytes at their final differentiation step[12,52]. Note that mitochondrial elimination occurs almost normally proceeds in *Atg5*KO erythrocytes, whereas it is largely disrupted in erythrocytes lacking Ulk1, which is a key molecule both in autophagy and in GOMED[1,4], indicating that this phenomenon is GOMED-dependent. This is supported by morphological data, because condensed mitochondria were engulfed by Golgi-like

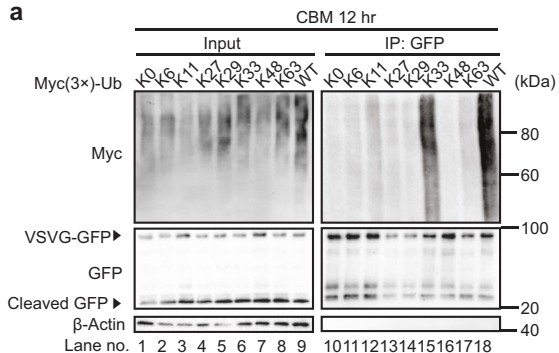

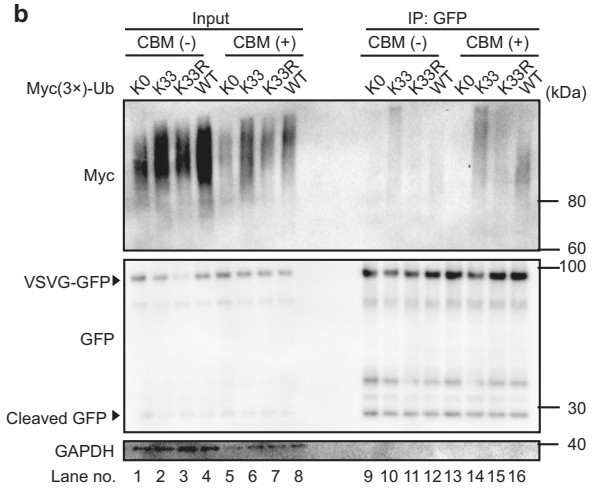

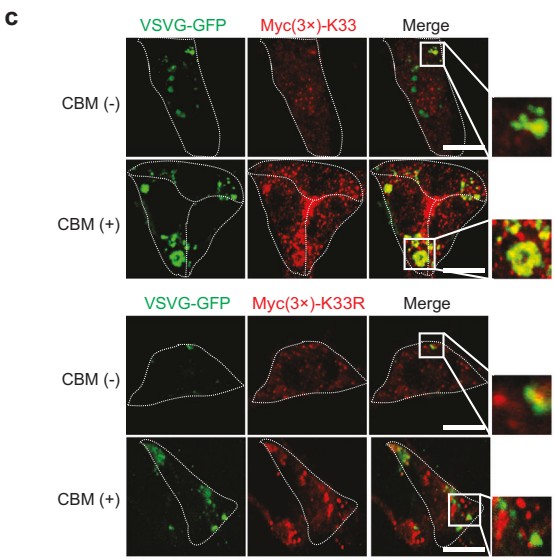

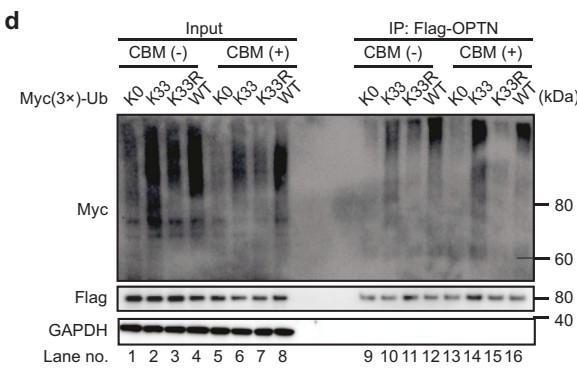

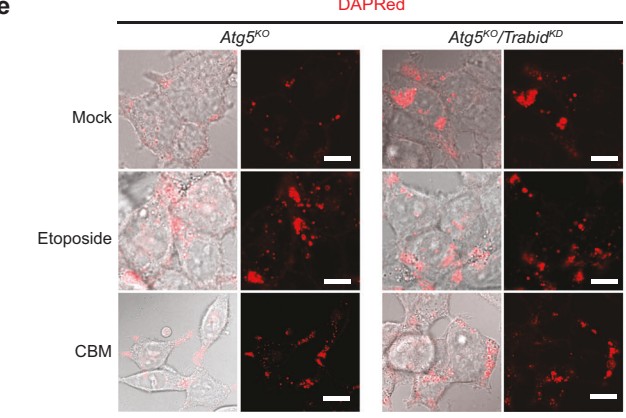

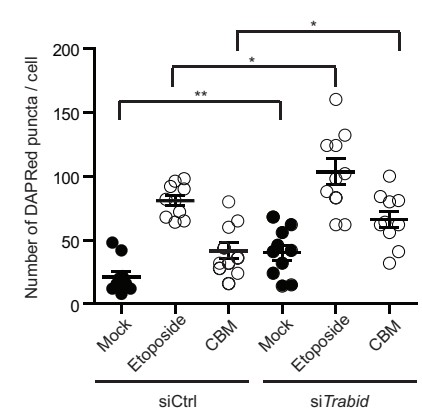

membranes in erythroblasts from wild-type (WT) and *Atg5*<sup>KO</sup> embryos (Fig. 8a, Supplementary Fig. 16). In contrast, in *Ulk1*<sup>KO</sup> erythrocytes, in which GOMED is inhibited at an early step, multiple mitochondria were attached to the extended Golgi membrane (Fig. 8a, Supplementary Fig. 16). This morphology was owing to a dysfunction of the formation of phagophore-like GOMED structures. The GOMED-dependent elimination of mitochondria was also confirmed by the colocalization of

Tom20 (a mitochondrial protein) with Lamp2, but not LC3, in Ter119⁺ cells (indicating erythrocytes) (Fig. 8b). These findings indicated that the removal of mitochondria from erythrocytes is carried out by GOMED. Therefore, we hypothesized that mitochondria might be ubiquitinated, recognized by Optn, and degraded by GOMED during erythrocyte maturation. In fact, when we collected mature erythrocytes from WT neonate mice as Ter119⁺ cells, there were few

**Fig. 5 | K33-polyUb is crucial for VSVG-GFP recognition by OPTN. a, b** K33-ubiquitination of VSVG-GFP upon CBM treatment. VSVG-GFP and various Myc (3×)-tagged Ub chains were transiently expressed in HEK293T cells. After 24 hr, cells were treated with or without CBM (1 mM) for 12 hr, and immunoprecipitated with an anti-GFP antibody. Expression of Myc (3×)-tagged Ub chains, VSVG-GFP/GFP, and GAPDH in total lysates (Input) and immunoprecipitants (IP: GFP) were analyzed by western blotting. The cleavage of VSVG-GFP is not substantially affected by the type of exogenous Ub chains, owing to the presence of endogenous Ub. Note that K33-Ub, but not K33R-Ub, was accumulated in total lysates and GFP immunoprecipitants. **c** Colocalization of K33-Ub chains with VSVG-GFP upon CBM treatment. The Myc (3×)-K33-Ub or Myc (3×)-K33R-Ub plasmid was transfected into VSVG-GFP-expressing $Atg5^{KO}$ MEFs. After 20 hr, cells were treated with CBM (1 mM) for 12 hr, immunostained with an anti-Myc antibody, and images of VSVG-GFP (green) and Myc (red) were obtained. Representative images are shown. Bars = 10 μm. ROIs are indicated by squares, and their magnified images are shown in the right panels. Note that Myc (3×)-K33-Ub, but not Myc (3×)-K33R-Ub, was colocalized with VSVG-GFP

upon CBM treatment. **d** Interaction of K33-Ub with OPTN upon CBM treatment. HEK293T cells were transfected with plasmids encoding Flag-OPTN and the indicated Myc (3×)-tagged Ub chains. After 24 hr, cells were treated with or without CBM (1 mM) for 12 hr, and immunoprecipitated with an anti-Flag antibody. Expression of Myc (3×)-tagged Ub chains, Flag-OPTN, and GAPDH in total lysates (Input) and immunoprecipitants (IP: Flag) were analyzed by western blotting. Note that K33-Ub, but not K33R-Ub, was coimmunoprecipitated with Flag-OPTN. **e, f** Crucial role of Trabid in GOMED. $Atg5^{KO}$ MEFs were transfected with siRNA against *Trabid* or a control siRNA. After 24 hr, DAPRed was added to the cells, followed by treatment with etoposide (10 μM) or CBM (1 mM) for 12 hr, and then DAPRed fluorescence images and phase-contrast images were obtained. In **e** representative images are shown. Bars = 10 μm. In **f** the number of DAPRed puncta per cell was calculated. Data are shown as the mean ± SE ($n$ = 10 cells, experiments were performed 2 times). Comparisons were performed using one-way ANOVA followed by the Tukey' *post-hoc* test. *$p$ < 0.05, **$p$ < 0.01. Data are representative of three independent experiments in (**a, b, d**) and two independent experiments in (**c, e**).

mitochondria, and several poly-Ub signals were observed in the cytosol (Fig. 8c). On the other hand, a substantial number of mitochondria remained in the mature erythrocytes from $Ulk1^{KO}$ neonate mice, and many Ub signals were located on the mitochondria (Fig. 8c). Optn also colocalized with Tom20, a molecule on the outer mitochondrial membrane, in $Ulk1^{KO}$ erythrocytes (Fig. 8d), suggesting the possibility that Optn accumulates on ubiquitinated mitochondria in $Ulk1^{KO}$ erythrocytes because of GOMED dysfunction. To test the involvement of Optn in mitochondrial elimination, we analyzed the amount of mitochondria in the erythrocytes of $Optn^{KO}$ neonate mice[53]. We found that mitochondria signals were absent in most erythrocytes from WT mice and $Optn^{heteroKO}$ mice, whereas they were observed in $Optn^{KO}$ erythrocytes (Fig. 8e–h). Not only did $Optn^{KO}$ erythrocytes have mitochondria, but they also demonstrated the deformation of erythrocyte morphology into a spiny-like shape (Fig. 8f). The abnormal morphology and presence of mitochondria in erythrocytes with GOMED dysfunction is consistent with our previous reports[12]. Taken together, our results demonstrate that Optn functions to eliminate mitochondria from maturing erythrocytes by recognizing Ub for GOMED degradation.

## Discussion

The mechanism of substrate recognition in GOMED has not yet been clarified to date. In this study, we found that GOMED recognizes and degrades substrate proteins selectively, similarly to selective autophagy. We also identified OPTN as an adaptor protein, which recognizes K33-polyubiquitinated substrates in GOMED. In vivo, this mechanism functions in the elimination of mitochondria from terminally differentiated erythrocytes. These results clarify how GOMED recognizes and degrades substrates, and demonstrates new functions for OPTN and K33 Ub chains.

Ub chains are known to play a variety of roles in addition to its role in proteolysis. In particular, the functions of K48, K63, and linear polyUb chains have been well studied. On the other hand, the functions of the remaining five atypical Ub chain types (linked via K6, K11, K27, K29, and K33) are less well defined[19–22]. Regarding the K33 Ub chain, its association with canonical autophagy has been pointed out[54], but its biological role has not yet been fully clarified, owing to its low intracellular amount[55]. In the present study, we demonstrated that the K33 Ub chain is used to mark substrate proteins for degradation by GOMED. This is evidenced by the fact that K33-Ub is added to the substrate molecules of GOMED, that K33-Ub increases when GOMED is inhibited, and that K33-polyUb can be detected in immunoprecipitants of OPTN. Furthermore, knockdown of *Trabid*, the DUB of K33-polyUb, increases the amount of K33-polyUb and enhances substrate degradation by GOMED, indicating that K33-polyUb functions in substrate recognition in GOMED. However, although it is clear that the K33 Ub chain is involved in substrate recognition, it is not clear whether this is

a simple K33 Ub chain or a Ub chain with branching or modifications, and further analysis is required. Importantly, it is known that K48-polyUb and K63-polyUb degrade proteins using proteasome and autophagy, respectively, and our findings indicate a third proteolytic machinery for Ub, namely GOMED, in which K33-polyUb is used for proteolysis.

In this study, we further identified OPTN as an adaptor molecule for GOMED that recognizes K33-polyUb. It is reasonable that OPTN, one of the five autophagy adaptor molecules reported to localize to the Golgi, is involved in GOMED, because the Golgi membrane is the membrane source of GOMED[4,6]. Recent studies have identified autophagy adaptors specific to Golgiphagy[56,57]; however, in the context of GOMED, OPTN acts as a key mediator. OPTN is known to play roles in vesicular trafficking, NF-κB signaling, and autophagy[30,37,58], and we here show the novel biological role of OPTN in GOMED. In autophagy, OPTN plays a role as an adaptor protein that traps ubiquitinated substrates, in which K63 chains rather than K48 chains preferentially interact with the UBD of OPTN. In NF-κB signaling, linear Ub chains interact with the UBD of OPTN[37]. On the other hand, we here found that OPTN selectively binds K33 chains at the ZF domain, when substrates are degraded by GOMED. Therefore, it appears that the difference in the type of Ub chain and difference in the domain of OPTN involved determine its particular cell function, including NF-κB activation, autophagy, and GOMED.

*OPTN* is a causative gene for Amyotrophic lateral sclerosis (ALS) and glaucoma[59–61], and in the case of ALS, the E478G mutation in *OPTN* is known to cause the disease[62]. Because this mutation is in the UBD, there are reports attributing this mutation to abnormal NF-kB signaling and autophagy[37]. However, recent GWAS studies have demonstrated that amino acid mutations within the ZF domain can also cause ALS. Specifically, the heterozygous missense mutations c.1690 G > C (p.D564H) and p.K557T have been found in patients with sporadic ALS[63,64], and these mutations are not reported in the healthy human database. Replacement of Asp564 to His results in an unstable ZF, suggesting that impaired substrate recognition in GOMED contributes to the onset of ALS[64]. The possibility that GOMED may be involved in ALS caused by abnormalities in the OPTN ZF domain is also supported by the finding that neuron-specific GOMED-deficient mice show neurodegenerative-like symptoms, and by the report that ALS is caused by abnormalities in the Ulk1 molecule, which is an important molecule in GOMED[65].

In addition to neurodegenerative-like symptoms, other abnormalities caused by GOMED dysfunction in mice include the disturbance of erythrocyte maturation, in which mitochondrial removal during erythroid differentiation is inhibited. This phenomenon has been observed in Ulk1-deficient mice, and has been demonstrated to be owing to GOMED dysfunction. Detailed observation of $Ulk1^{KO}$ erythrocytes using electron microscopy demonstrated that the residual

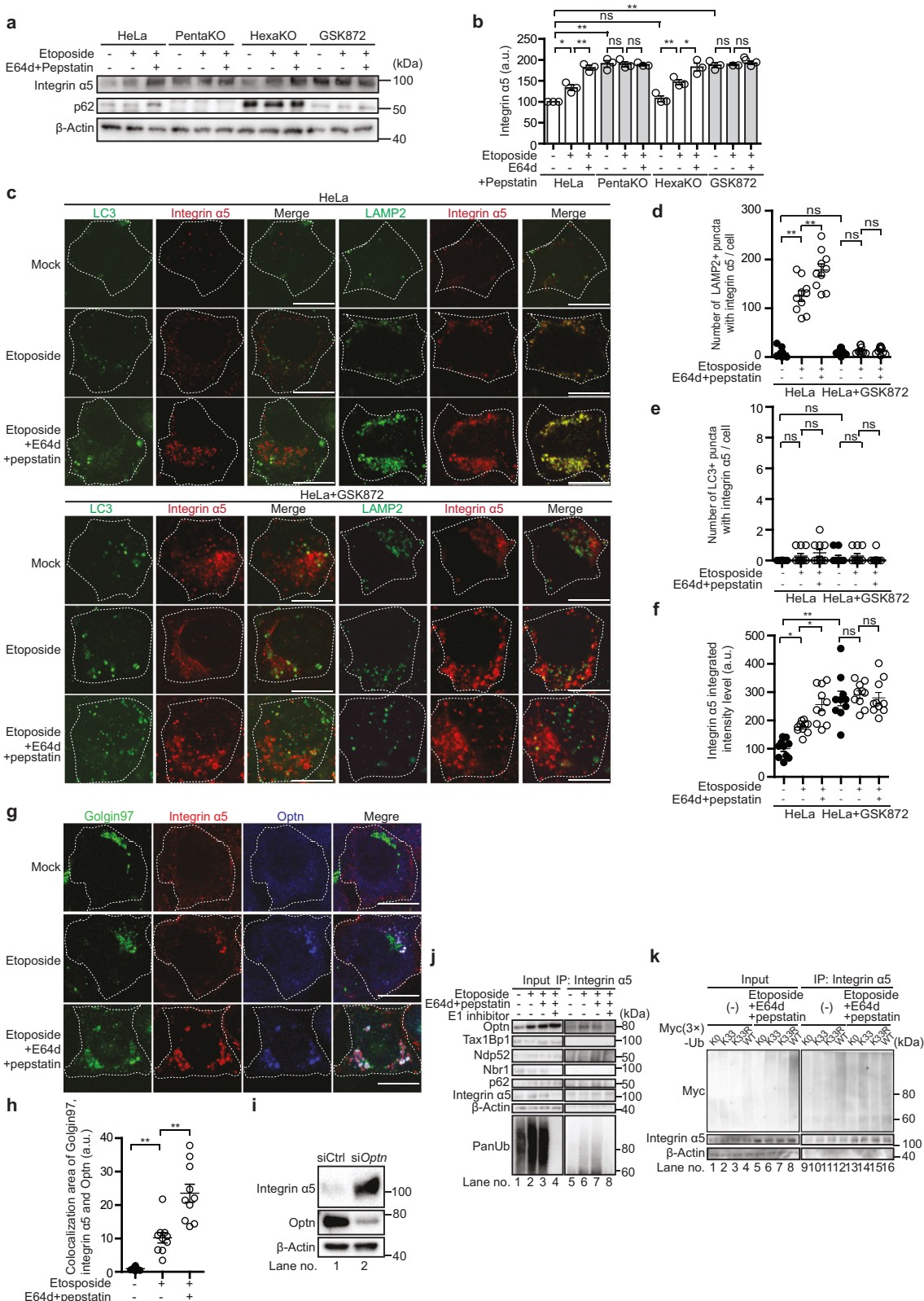

mitochondria were not engulfed by, but were in contact with the Golgi membrane[4,12]. In *Optn*[KO] mice, mitochondria elimination from erythrocytes was not expected to occur owing to a dysfunction of substrate recognition, and in fact, multiple mitochondria were found to be present in *Optn*[KO] erythrocytes. In addition, *Ulk1*[KO] erythrocytes showed morphological abnormalities in association with residual mitochondria, and a similar morphology was observed in *Optn*[KO]

erythrocytes. These results indicate that OPTN is involved in GOMED substrate recognition in vivo.

## Methods

### Cell lines
Human embryonic kidney (HEK293T) cells were cultured in Dulbecco's modified Eagle's medium supplemented with 10% fetal bovine serum

**Fig. 6 | Endogenous GOMED substrate integrin α5 is degraded via the K33-polyUb-OPTN pathway. a, b** HeLa cells, PentaKO cells, HexaKO cells, and HeLa cells incubated with GSK872 (10 μM) were treated with etoposide (10 μM) with or without E64d (10 μg/mL) / pepstatin (10 μg/mL) for 12 hr. Cells were then lysed, and the expression of each protein was analyzed by western blotting. p62 was used as a marker of autophagy; PentaKO cells lack p62, and HexaKO cells are deficient in autophagy. β-actin was used as a loading control. **b** Semiquantitative analysis of protein expression in the experiment in (**a**). Data are shown as the mean ± SE (*n* = 3). **c–f** HeLa cells and HeLa cells incubated with GSK872 (10 μM) were treated with etoposide (10 μM) with or without E64d (10 μg/mL) / pepstatin (10 μg/mL) for 12 hr. Cells were immunostained with anti-LC3 (green), anti-integrin α5 (red), and anti-LAMP2 (green) antibodies. In **c** representative images are shown. White dotted lines indicate the cell shapes. Bars = 10 μm. The number of integrin α5 puncta colocalizing with LAMP2 (**d**) and with LC3 (**e**) among total integrin α5, and integrin α5 integrated intensity (**f**) were analyzed using ImageJ software. Data are shown as the mean ± SE (*n* = 10 cells). Note that several z-stack images were taken, and quantification was performed on a per-cell basis. **g–i** *Atg5*^KO MEFs were treated with etoposide (10 μM) together with or without E64d (10 μg/mL) / pepstatin (10 μg/mL) for 12 hr. Cells were then immunostained with anti-Optn (blue), anti-integrin α5 (red), and anti-Golgin97 (green) (**g, h**) antibodies. Representative images are shown in (**g**). Bars = 10 μm. In (**h**), the area of colocalization between integrin α5 and Optn on Golgin97 was calculated using ImageJ software. Data are shown as the mean ± SE

(*n* = 10 cells, experiments were performed two times). Etoposide increased the colocalization of integrin α5, Optn, and Golgi or lysosomes, which was further increased by the inhibition GOMED by E64d/pepstatin. **i.** Requirement of Optn for integrin α5 degradation. *Optn* was knocked down by siRNA in *Atg5*^KO MEFs. After 24 hr, the expression of each protein was analyzed by western blotting. β-actin was used as a loading control. **j.** Interaction of ubiquitinated integrin α5 with Optn. Similar experiments to Fig. 4b were performed using etoposide-treated *Atg5*^KO MEFs, and endogenous integrin α5, instead of exogenous VSVG-GFP, was analyzed. In this experiment, cells were treated with or without TAK-243 (100 nM) for 12 hr before cell collection. Consistent with VSVG-GFP, Optn was coimmunoprecipitated with integrin α5 upon etoposide treatment. **k** HEK293T cells were transiently expressed with various Myc (3×)-tagged Ub chains. After 24 hr, cells were treated with etoposide (10 μM) in the presence of E64d (10 μg/mL) / pepstatin (10 μg/mL) for 12 hr or left untreated, and then immunoprecipitated with an anti-integrin α5 antibody. Expression of Myc (3×)-tagged Ub chains, integrin α5, and β-actin in total lysates (Input) and immunoprecipitants (IP: integrin α5) was analyzed by western blotting. Note that K33-Ub, but not K33R-Ub, was accumulated in total lysates and immunoprecipitants of integrin α5. In **b–f, h, i** comparisons were performed using one-way ANOVA followed by the Tukey's *post-hoc* test. **p* < 0.05, ***p* < 0.01; ns: not significant. Data are representative of three independent experiments in (**a, j** and **k**) and two independent experiments in (**c, g** and **i**).

---

and 1% penicillin-streptomycin at 37 °C. WT, *Atg5*^KO, and R26-Golgi-mCherry/*Atg5*^KO MEFs were previously generated by the authors from mouse embryos at embryonic day 13.5[4,39]. The generation of VSVG-GFP-expressing *Atg5*^KO MEFs, PentaKO and HexaKO HeLa cell lines were also described previously[6,34]. *Optn*^KO MEFs were originally generated from *Optn*^KO embryos (E14.5). MEFs were cultured in Dulbecco's modified Eagle's medium supplemented with 2 mM L-glutamine, 1 mM sodium pyruvate, 0.1 mM nonessential amino acids, 10 mM HEPES/Na+ (pH 7.4), 0.05 mM 2-mercaptoethanol, 1% penicillin-streptomycin and 10% fetal bovine serum.

### Animals
C57BL/6 mice were purchased from Japan CLEA (Tokyo, Japan). The generation of *Ulk1*^KO mice and *Optn*^KO mice has been described previously[12,53]. Mice were bred on a 12-h light/12-h dark cycle at approximately 23 °C and 40% relative humidity at the Laboratory for Recombinant Animals of Institute of Science Tokyo, Tokyo, Japan. This animal facility is operated according to the NIH guidelines. The Institute of Science Tokyo Ethics Committee for Animal Experiments approved all experiments in this study, and all experiments were performed according to their regulations.

### Antibodies and chemicals
The antibodies and chemicals used are listed in Supplementary Table 1.

### RNA interference
ONTARGETplus SMARTpool siRNA oligonucleotides specific for mouse *Optn* and *Trabid* and nontargeting siRNA (control siRNA) were purchased from Dharmacon. ON-TARGETplus Mouse Optn (71648) siRNA (Set of 4; four oligonucleotides specific for mouse *Optn*) was also purchased from Dharmacon. Cells were transfected with 20 nM siRNA against human *OPTN* or mouse *Optn* using Lipofectamine RNAiMAX. The siRNAs in Set Of 4 were each transfected one by one.

### Immunoblotting
Cells were incubated in lysis buffer (20 mM Tris–HCl [pH 7.5], 150 mM NaCl, 0.2% Nonidet P-40, 10% glycerol, 2 mM N-ethylmaleimide, protease inhibitors, and Halt protease and phosphatase inhibitor cocktail) on ice for 20 min, and centrifuged at 15,000 × *g* for 20 min. For Flag or GFP immunoprecipitation, cell lysates were incubated with anti-DDDDK-tag mAb-magnetic beads or anti-GFP-tag mAb-magnetic beads. For VSVG, integrin α5 and MSLN

immunoprecipitation, anti-VSVG, integrin α5 and MSLN Abs were coupled with Dynabeads protein G for 1 hr at 4 °C. Cell lysates were incubated with the precoupled beads for 1 hr at 4 °C. Samples were resolved on NuPage precast 4% to 12% Bis-Tris gels, and transferred to polyvinylidene difluoride membranes. Membranes with proteins were blocked with 5% skim milk in TBS-T, and incubated with the indicated primary antibodies overnight at 4 °C. After washing three times with TBS-T, the membranes were incubated with a horseradish peroxidase-labeled secondary antibody, and visualized with Chemi-Lumi One Super reagent. The antibodies used in this study are listed in Supplementary Table 1.

### Denatured immunoprecipitation analysis
For denatured immunoprecipitation analysis, the denaturing SDS lysis solution was prepared by resuspending the cell pellet in 100 μL of a solution containing 2% SDS and 5 mM DTT. The sample was heated at 95 °C for 5 min, then cooled to room temperature and vigorously mixed. Next, the sample was diluted with Nonidet P-40 buffer so that the final concentrations of the lysis solution used for immunoprecipitation was 0.2% SDS, 0.5 mM DTT, 0.5% Igepal, 50 mM Tris (pH 8.0), 150 mM NaCl, and 10 mM MgCl$_2$. Cell lysates were incubated with the anti-GFP-tag mAb-magnetic beads for 1 hr at 4 °C.

### Plasmids
The plasmids and primers used in this study are listed in Supplementary Table 2 and Supplementary Table 3. Human *OPTN*, *NBR1*, *NDP52*, and *TAX1BP1* human Ub cDNA were cloned into pCMV-3Tag-2C plasmids. The Ub mutants, K0, K33, and K33R were described previously[54], and K6, K11, K27, K29, K48, and K63 were prepared using the PrimeSTAR mutagenesis basal kit. Flag-OPTN-ΔZF and ΔUBA were generated using the PrimeSTAR mutagenesis basal kit. All constructs were confirmed by sequence analysis. To create siRNA-resistant Flag-OPTN, primers that introduce nucleotide mutations without causing amino acid changes were designed based on the sequence provided by Dharmacon. Using these primers and the PrimeStar Mutagenesis Basal Kit, mutations were introduced into the Flag-Optn plasmid to generate siRNA-resistant Flag-Optn (R1-4). The primers used are listed in KEY RESOURCES TABLE. HEK293T and PentaKO cells were transiently transfected with these plasmids using Lipofectamine 2000, as indicated by the manufacturer. MEFs were transiently transfected using 4D-Nucleofector X Unit and P4 Primary Cell 4D Nucleofector X Kit L as recommended by the manufacturer.

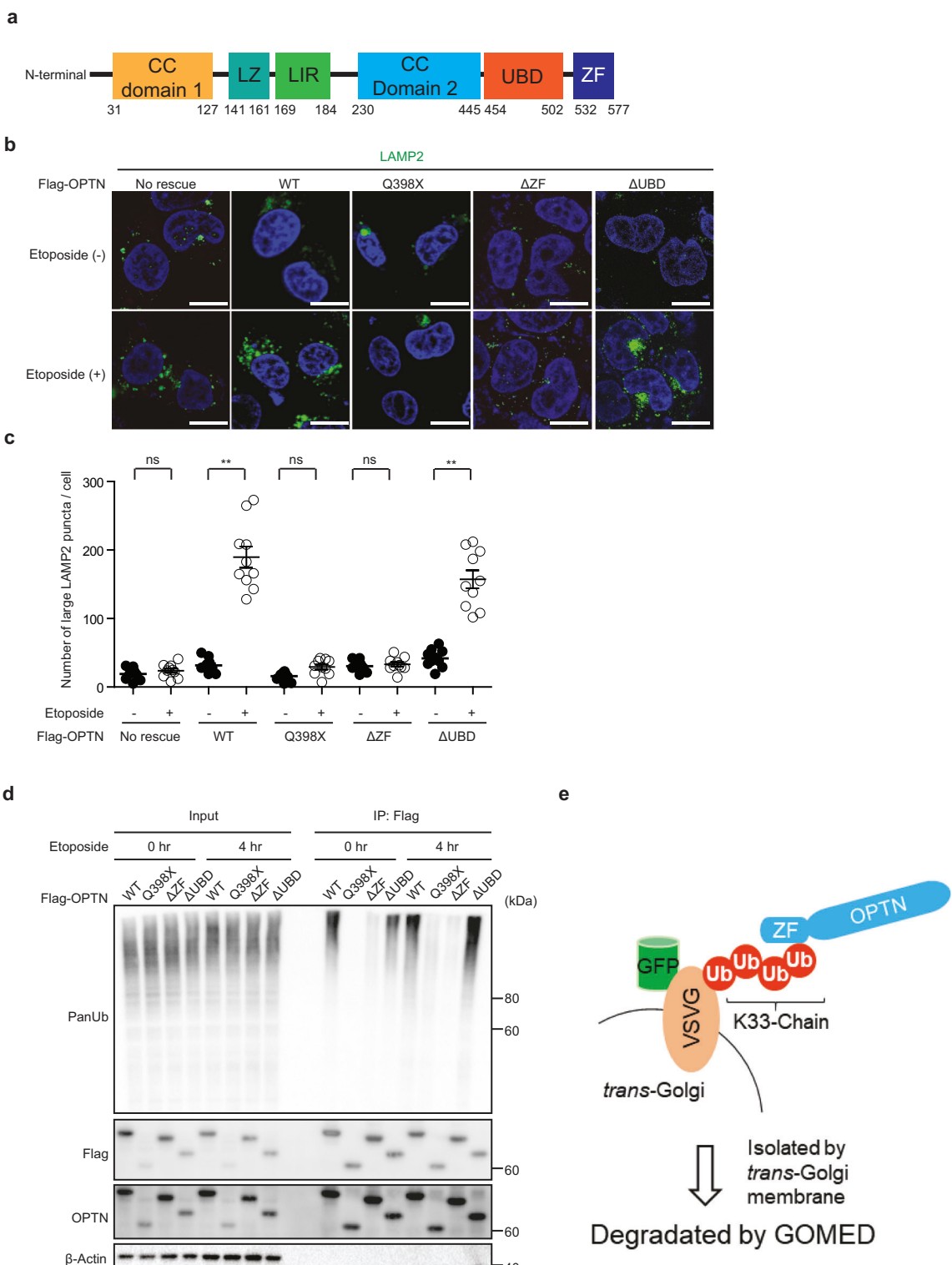

**Fig. 7 | OPTN recognizes polyUb via its ZF domain in GOMED. a** Schematic diagram of the structure of OPTN. Amino acid numbers are shown at the bottom. UBD and ZF are Ub-binding domains. **b, c** Crucial role of the ZF domain in etoposide-induced GOMED. PentaKO HeLa cells were transfected with plasmids encoding Flag-tagged-OPTN and its mutant (Q398X, ΔZF, and ΔUBD) plasmids. After 24 hr, cells were treated with etoposide (10 μM) for 12 hr, and immunostained with an anti-LAMP2 antibody. Nuclei were counterstained with DAPI. In **b** representative images are shown. Bars = 10 μm. In **c** the number of large LAMP2 puncta per cell was calculated. Data are shown as the mean ± SE ($n = 10$ cells, experiments were performed two times). Comparisons were performed using one-way ANOVA followed by the Tukey's *post-hoc* test. \*\*$p < 0.01$; ns: not significant. **d** No interaction of polyUb with ZF domain lacking OPTN upon etoposide treatment. HEK293T cells were transiently transfected with either a plasmid encoding Flag-tagged-OPTN or its mutants (Q398X, ΔZF, and ΔUBD). After 24 hr, cells were treated with or without etoposide (10 μM) for 4 hr, and immunoprecipitated with an anti-Flag antibody. Expression of polyUb, Flag-OPTN mutants, and β-actin in total lysates (Input) and immunoprecipitants (IP: Flag) were analyzed by western blotting. Note that polyUb was not coimmunoprecipitated with Optn lacking the ZF domain. **e** Schematic diagram showing that OPTN interacts with K33-ubiquitinated substrates via its ZF domain, with its substrates subsequently being degraded by GOMED. This scheme was designed and created by the authors using Microsoft PowerPoint 365. Data are representative of two independent experiments in (**b, d**).

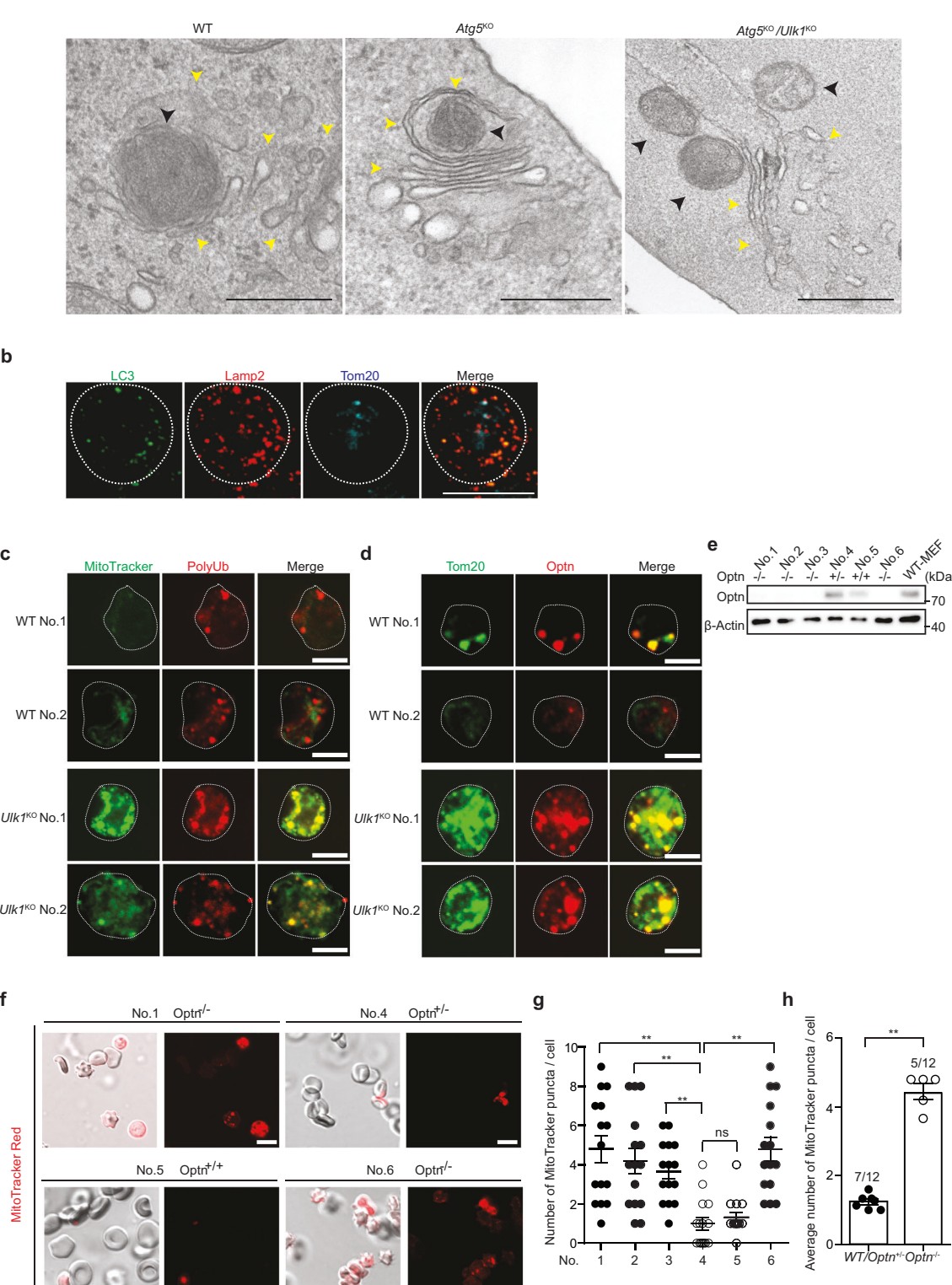

### Autophagy detection by DAPGreen and DAPRed staining

The regents used in this study are listed in Supplementary Table 4. Cells were stained with 0.1 μM DAPGreen or DAPRed working solutions, as previously reported[39]. After 30 min, the cells were washed with PBS. Subsequently, cells were treated with various stimuli, and nuclei were counterstained with 1 μg/mL Hoechst 33342. Images were acquired using a confocal laser-scanning microscope (FV10i or FV3000, Olympus) with a 60× or 100× oil-immersion objective lens.

### Immunofluorescence analysis

Mitochondria were stained using MitoTracker Green and Red FM as stated in the manufacturer's instructions. Cells were washed with PBS, and fixed in 4% paraformaldehyde for 10 min at 4 °C. Fixed cells were

**Fig. 8 | Requirement of Optn for mitochondria elimination during erythroid maturation. a** Representative EM images of reticulocytes obtained from the livers of WT, *Atg5^KO*, and *Atg5^KO*/*Ulk1^KO* mouse embryos (embryonic day 14.5). Black arrowheads and yellow arrowheads indicate mitochondria and Golgi, respectively. Slightly condensed mitochondria were engulfed by double membranes next to the Golgi apparatus in WT and *Atg5^KO* reticulocytes (bar = 0.5 μm). In contrast in *Atg5^KO*/*Ulk1^KO* reticulocytes, multiple mitochondria were attached to the straightly elongated Golgi membrane (bar = 0.5 μm). **b** Peripheral blood was collected from WT mice on postnatal day 1, and erythrocytes were collected using an anti-Ter119 antibody. Erythrocytes were stained with anti-LC3 (autophagy marker; green), anti-Tom20 (mitochondria marker; blue) and anti-Lamp2 (red) antibodies. White lines indicate the cell shapes. Bar = 5 μm. Tom20 and Lamp2, but not LC3, were merged. **c** PolyUb accumulation on residual mitochondria in *Ulk1^KO* erythrocytes. Peripheral blood was collected from WT and *Ulk1^KO* mice on postnatal day 1, and erythrocytes were collected using an anti-Ter119 antibody. Then, erythrocytes were stained with MitoTracker Green, and immunostained with an anti-polyUb antibody. Representative images are shown. Bars = 5 μm. **d** Localization of Optn on residual mitochondria in Ulk1-deficient erythrocytes. Similar experiments with (**c**) were performed, followed by immunostaining with anti-Tom20 (green) and anti-Optn (red) antibodies, instead of an anti-polyUb antibody. Representative images are shown. Bars = 5 μm. **e** Confirmation of *Optn* genotype and protein expression in neonatal mice. Tails were cut from neonatal mice (postnatal day 1) obtained by mating *Optn* heterozygous mice, and genotyped and analyzed for protein expression. Protein extract from wild-type MEFs was used as a positive control. **f–h** Lack of mitochondrial elimination from Optn-deficient erythrocytes. In **f, g** erythrocytes from the same mice as those in (**e**) were analyzed. Erythrocytes were obtained using an anti-Ter119 antibody, stained with MitoTracker Red, and observed using phase-contrast and fluorescence microscopy. In **f** representative images are shown. Bars = 10 μm. In **g** the number of MitoTracker Red puncta per erythrocyte was calculated. Data are shown as the mean ± SE (n = 15 cells). In **h** 12 neonatal mice (from two sets of parents) were analyzed, and the average number of MitoTracker Red puncta was counted. Data are shown as the mean ± SE. Comparisons were performed using one-way ANOVA followed by the Tukey's *post-hoc* test. **$p < 0.01$; ns: not significant. Data are representative of three two independent experiments in (**a–e, f**).

permeabilized with 50 μg/mL digitonin in PBS for 5 min, blocked with 3% bovine serum albumin in PBS for 15 min, and incubated with primary antibodies at 4 °C overnight. After washing, cells were incubated with Alexa Fluor 488-, 594-, or 647-conjugated anti-mouse/rabbit antibody for 30 min. In live-cell fluorescence imaging, the nuclei were counterstained with DAPI. Images were acquired using a confocal laser-scanning microscope (FV10i or FV3000, Olympus) with a 60× or 100x oil-immersion objective lens.

### Erythrocyte isolation
Ter119⁺ erythrocytes were isolated from the peripheral blood at postnatal day 1, using magnetic beads conjugated with an anti-Ter119 antibody (BD IMag Cell Separation System).

### Quantification and statistical analysis
Results are expressed as the mean ± standard error (SE). Statistical analyses were performed using Prism5 (GraphPad) software. Significance of differences between two selected groups were analyzed by the unpaired two-tailed Student *t*-test. For three or more samples, comparisons were performed using one-way ANOVA followed by the Tukey's *post-hoc* test (* $p < 0.05$, ** $p < 0.01$, and ns: not significant, in all experiments). Triple-color immunofluorescence images were analyzed using ImageJ software (Fuji). Specifically, fluorescence channels were split and binarized by manual thresholding to isolate specific signals. Logical AND operations were applied using the "Image Calculator" function to identify regions positive for all three markers. The area of triple-colocalized regions was quantified using the "Analyze Particles" function, and results were obtained from the summary output. Integrated fluorescence intensities were measured using ImageJ, by defining regios of interest (ROIs) and using the "Measure" function to obtain integrated densities. The background signal was determined using regions adjacent to the ROIs, and subtracted from the data. Colocalization between two fluorescence signals was quantified using the Pearson's correlation coefficient in ImageJ. After channel splitting and background subtraction, the "Coloc 2" plugin was used to calculate Pearson's correlation coefficients within defined ROIs. Results were based on at least 10 cells per condition.

### Electron microscopy
Cells were fixed with 1.5% paraformaldehyde/3% glutaraldehyde in 0.1 M phosphate buffer (pH 7.3), followed by treatment with 1% OsO4. After dehydration, the fixed cells were embedded in Epon812. Thin sections were cut and stained with uranyl acetate lead citrate for observation using a JEM-1400Plus electron microscope (JEOL Co. Ltd) at 100 kV.

### Correlative light and electron microscopy (CLEM)
MEFs cultured on glass-bottom dishes with grids were stained with DAPGreen and DAPRed and then treated with etoposide to induce GOMED. Samples were fixed with paraformaldehyde (0.75%)/glutaraldehyde (1.5%) in 0.1 M phosphate buffer (PB) at pH 7.4 at room temperature, and signals were observed using an LSM710 fluorescence microscope. Subsequently, the samples were incubated on ice for 30 min. The samples were then fixed with 2% glutaraldehyde in 0.1 M PB at 4 °C overnight. After fixation, the samples were washed three times with 0.1 M PB and postfixed with 1% OsO4 in 0.1 M PB at 4 °C for 1 hr. After dehydration, ultrathin sections were stained with 2% uranyl acetate and lead stain solution (Sigma-Aldrich), and observed using a JEM-1400Plus electron microscope (JEOL Co. Ltd) at 100 kV.

### Reporting summary
Further information on research design is available in the Nature Portfolio Reporting Summary linked to this article.

## Data availability
All data that supporting the findings of this study are available from the corresponding author upon request. Source data are provided with this paper.

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

## Acknowledgements

This study was supported by JSPS KAKENHI Grants-in-Aid for Scientific Research (S) (23H05480), (A) (20H00467), and (C) (20K07353, 24K11166, 22K07345, and 23K05748), and Grant-in-Aid for Encouragement of Young Scientists (B) (19K16119), and by MEXT KAKENHI Grants-in-Aid for Scientific Research on Innovative Areas (22H04639, 23H04773, and 24H01887). This study was also supported by AMED-CREST (JP23gm1410012), and by AMED under Grant Number JP21wm0525028, JP24ama221130, JP24ek0109770. This study was also supported by the Joint Usage/Research Program of Medical Research Institute, Institute of Science Tokyo. ML was supported by the National Health and Medical Research Council (NHMRC) (GNT1106471), the Australian Research Council Discovery Project (DP200100347) and the Rebecca Cooper Foundation Fellowship (RC20241396).

## Author contributions

S.S. conceived and supervised the project. Y.S. conducted all the experiments and analyzed the results with support from H.Y., S.H., H.S., S.T., S.A., M.L., and H.K. The work was carried out under the supervision of S.O., R.O., and S.S. Y.S. and S.S. wrote the manuscript. All authors read and approved the final version of manuscript.

## Competing interests

The authors declare no competing interests.
