## [Transparent Peer review file · Nature Communications]

Optineurin is an adaptor protein for ubiquitinated substrates in Golgi membrane-associated degradation

Corresponding Author: Professor Shigeomi Shimizu

Version 0:

Reviewer comments:

Reviewer #1

(Remarks to the Author)

The manuscript describes that 'Optineurin is an adaptor protein for ubiquitinated substrates in Golgi membrane-associated degradation'. Golgi membrane-associated degradation (GOMED) is an autophagy independent processes that leads to the degradation of proteins that have passed through the trans-Golgi membranes upon stress. This manuscript aims to explore how GOMED substrates are selected for degradation. The authors use genetics, imaging and biochemical approaches to arrive at the conclusion that – in response to various stressors - Optineurin detects K33 polyubiquitylated proteins at the Golgi for GOMED. While this is an interesting and important concept, the manuscript in its current form, unfortunately falls short to provide compelling evidence to support the concepts. A major revision, substantial improvement of data quality and additional data would be required

Major points:

- 1) What are the endogenous substrates of the OPTN-K33-GOMED pathway?
- 2) In many experiments it is difficult to directly discriminate GOMED and autophagy, due to the lack of an endogenous GOMED substrate.
- 3) The Data quality for some of the Western Blots experiments is suboptimal and (e.g.: Fig.3a, Fig.7c).
- 4) Sometimes it is difficult to match the presented WB with the quantification (e.g. Fig. 1a, b). The WB clearly shows upregulation of p62 and NDP52 and OPTN protein levels (lanes 4,5,6), but in the quantification on OPTN is upregulated and p62 is even downregulated.
- 5) The knock efficiency of OPTN appears to be rather moderate (Figure S2). OPTN knockdown experiments should be controlled by corresponding rescue experiments (eg. Figure 1c)
- 6) What is the cause for the difference in LC3 and DAPGreen punctae in p62 and NDP52 cells (Figure 2a,b,c)? does it imply that in addition to OPTN, also NBR1 and TXA1BP1 are GOMED receptors? In addition, the WB in Figure S4 (which should serve as a control for Figure 2a-c) lacks a critical control which makes it impossible to evaluate the protein expression levels of the adaptor proteins compared to the parental WT cells.
- 7) Figure 2a: Does the DAPGreen in the OPTN rescued pentaKO cells co-localize with lysosomal markers?
- 8) According to material and methods, the immunoprecipitation (IP) experiment used to show VSVG-GFP ubiquitylation were performed under 'native' conditions, such that interactions with other proteins would be retained. Hence proteins that interact with VSVG-GFP (or GFP fragments) could also be ubiquitylated. Moreover, the IP's lack controls (e.g. GFP only and untagged VSVG) and the use of the E1 inhibitor is not sufficient.
- 9) Based on the model propose VSVG-GFP should be degraded upon CBM treatment. Yet the fraction of VSVG-GFP that appears to be degraded is small (Figure 4e compare full length VSVG-GFP to the clipped/released GFP). Is the GFP release from full length VSVG-GFP dependent of the catabolic activity of lysosomes? Moreover, the bulk of VSVG-GFP accumulated at the Golgi (Figure 4d). How does that fit to the model?
- 10) Figure 5a: It seems that the expression of all UB chains triggers VSVG-GFP degradation (except for K0 UB, input lane 1)
- 11) The ratio of full length VSVG-GFP to clipped GFP differs between experiments. What could be the reason for these substantial differences?
- 12) Figure S5: K63 polyubiquitin chains were also co-IP with OPTN -> these are the autophagy cargoes?
- 13) The WB in Figure S6 is suboptimal – there are no bands visible for full length VSVG-GFP
- 14) Figure 7d,e: loss of OPTN could also cause mitophagy defects. It is impossible to discriminate the role of OPTN in autophagy or GOMED.

Minor points:

Figure 3C: The DAPI staining appears suboptimal.

Figure 3e: Please explain why LAMP2 is upregulated upon GOMED induction – are GOMED substrates sorted into these LAMP2 lysosomes?

Reviewer #2

(Remarks to the Author)

In this manuscript the authors set out to identify a substrate recognition mechanism of GOMED (Golgi membrane-associated degradation). GOMED is defined as a process that targets proteins that have passed through the Golgi for degradation (however, mitochondria are introduced as an additional substrate). As further stated in the manuscript, GOMED functions by generating autophagosome-like structures, whose formation does not depend on ATG9a, ATG5/7, or WIPI1/2 proteins. Using VSVG-GFP, a previously established GOMED substrate the authors identify a role for Optineurin (OPTN), an autophagy receptor, as a substrate receptor for GOMED. They propose GOMED to be a ubiquitin-dependent pathway. Their data show that OPTN interacts with K33-linked Ub chains and that the ZF domain of OPTN is required for this interaction. The dependence of VSVG-GFP degradation (as reflected by the accumulation of free GFP) is dependent on OPTN in ATG5 KO cells. The authors further propose an OPTN-GOMED axis to be important for the degradation of mitochondria during erythroid maturation.

While this manuscript provides insights into the role of OPTN in ubiquitin-mediated VSVG-EGFP degradation and in the degradation of mitochondria, more evidence is needed to substantiate the proposed model linking GOMED, OPTN, and the two very distinct substrates (VSVG-GFP, passing through the Golgi and mitochondria).

Major points:

1) The authors address the ubiquitination and degradation of the GOMED substrate VSVG-GFP (Figure 4). From the experiment shown in Figure 4a and b it cannot be concluded that VSVG is directly ubiquitinated. A denaturing IP needs to be performed to exclude that the poly-Ub signal stems from VSVG-GFP interaction partners. Does the VSVG-GFP construct contain lysine residues in its cytosolic tail? Lines 195-201 – Minor: The cartoon in Figure 4g should be updated for the GFP domain facing the Golgi lumen.

A quantification for the imaging data in Figure 4d should be included to assess the co-localization of OPTN and VSVG-GFP at the Golgi.

2) In the final Figure 7, the involvement of OPTN in the elimination of mitochondria from erythrocytes was addressed. This process has previously been shown to depend on Ulk1, an enzyme that is part of the ULK autophagy initiation complex. A knock-out of Ulk1 may impair GOMED (line 252) but in addition it also impairs classical autophagy and the accumulation of OPTN on mitochondria can therefore not be directly related to as dysfunction in GOMED.

In general, OPTN is not an exclusive substrate receptor for GOMED but also interacts with LC3 proteins (LIR domain of OPTN) and functions in the classical autophagy pathway. OPTN has previously been shown to associate for example with damaged mitochondria and to target them for autophagic degradation. 10.1073/pnas.1405752111

The involvement of OPTN does not equal the involvement of GOMED. It can therefore not be assumed that phenotypes observed in OPTN KO cells are always related to the impairment of GOMED. Is there any evidence that Golgi membranes are forming autophagosome-like structures around mitochondria?

Figure 7c – given the strong difference in ActB between WT and the Optn KO samples in conjunction with the weak signal for OPTN in WT-MEFs, it is difficult to assess any potential reduction in the OPTN levels/complete depletion of OPTN in No. 1-6.

3) Figure 1 – the quantification in panel b does not reflect the example image in panel a. Specifically the levels of OPTN strongly decrease (comparing 0 and 12h) based on the WB, but do not change according to the bar graph shown in b. Minor: The y-axis of all bar graphs is labeled as arbitrary units. Given that all zero-time point data points show 100 AU, the data are likely normalized and should be labelled as such.

4) Related to (3) – Figure 3ab – which show similar data as in Figure 1ab with a different treatment – The levels of p62, Nbr1, and Ndp52 are barely detectable in the example images. The levels of TAX1BP1 also decrease upon CBM treatment in ATG5 KO cells. Can the authors comment on the potential involvement of TAX1BP1?

5) The authors define DAPGreen+/LC3- puncta as GOMED puncta. The co-localization of these puncta with a GOMED marker needs to be included. The LC3- negative puncta may be positive for another mammalian ATG8-type protein and relate to classical autophagy pathways. Structures defined as GOMED puncta need to be clearly distinguished from classical autophagosomes.

Minor points:

The introduction falls a bit short in its description of autophagy and ubiquitin signaling.

For example: ...autophagy mainly degrades cytosolic proteins, whereas GOMED mainly degrades molecules that have passed through the Golgi. – The substrate spectrum of autophagy is much broader (including cellular organelles (mitochondria, lysosomes, etc), intracellular pathogenic organisms) and should also be reported as such.

The Ub code is more complex than reported in lines 80-83. More than K48, K63 and linear ubiquitin chains were shown to

induce proteasomal degradation or autophagy (lines 93-95). Given the topic and conclusion of the paper, please add a more detailed introduction on the role/function of different Ub chains.

How were the time points chosen for experiments shown in 1e and g? Based on WB in panel a, OPTN is largely degraded at the 12h time point. A time course experiment or additional analysis of an earlier time point would strengthen the argument that OPTN localizes to Golgi membranes upon etoposide treatment.

Can the authors comment on why GFP is not degraded by GOMED, but remains as a degradation product from VSVG-GFP degradation in lysosomes?

The presentation of the experiments performed with Ubiquitin mutants lacks clarity. From the text on page 8, it is not clear which type of Ub mutants were used – presumably Ubiquitin which exclusively harbors one lysine residue – K33? Please clarify.

Version 1:

Reviewer comments:

Reviewer #1

(Remarks to the Author)

In my view, the revised manuscript unfortunately does not provide compelling evidence that endogenous Golgi substrates for GOMED are K33-ubiquitinated and subsequently targeted into lysosomes by Optn.

Specifically, the experiments using integrin $\alpha 5$ as an endogenous substrate (presented in Figures S11 and 6) for GOMED are not particularly convincing for the following reasons:

1. Immunofluorescence (IF) staining alone does not provide sufficient evidence to draw conclusions regarding the protein expression levels of integrin $\alpha 5$.
2. There is clear co-localization of integrin $\alpha 5$ with LC3. Of course this is less pronounced compared to LAMP1, due to the relatively lower abundance of LC3 dots.
3. Curiously, Optn appears to constitutively interact with integrin $\alpha 5$ even in the absence of etoposide treatment. Furthermore, this interaction seems to be largely independent of ubiquitination, as it persists - albeit at a slightly reduced level - even after E1 inhibition, which effectively abrogates integrin $\alpha 5$ ubiquitination.

Reviewer #2

(Remarks to the Author)

In the revised version the authors present additional data to substantiate their findings. The following concerns regarding the data and conclusions remain:

- 1) Data on integrin $\alpha 5$ was added to analyze the substrate-targeting mechanism of GOMED i.e. OPTN in the context of an endogenous substrate.
 - a. The authors refer to changes in integrin expression levels – is there evidence that the expression of integrin $\alpha 5$ changes? If not, please remove the term 'expression'.
 - b. Panel c suggests that the levels of integrin $\alpha 5$ change upon etoposide treatment. In panel k however, the Western blot data show that the levels of integrin $\alpha 5$ do not change. How can these data be reconciled? Could the effect observed in panel a be due to changes in localization/clustering rather than a change in levels?
 - c. Why was the fluorescence intensity x area/cell quantified (assuming mean fluorescence intensity?) rather than the integrated intensity per cell (panel 6c and SI)?
 - d. The description in lines 319-312 does not fully match the data shown in Fig 6c. Based on the quantification the levels of integrin $\alpha 5$ increase in the PentaKO cells without any treatment. The relative increase produced by Etoposide and E64d+pepstatin treatment resembles the effect observed in WT cells.
 - e. Panel j represents a strong result for Optn involvement in integrin $\alpha 5$ turnover (while this does not equal an involvement of GOMED).
- 2) How were the imaging data quantified? I could not identify a detailed methods section on this aspect.
 - a. In Fig 1f and h, puncta were quantified, though I could not identify any puncta in the images.
 - b. In Fig 6h Golgin97 puncta were quantified – In mock-treated cells, the Golgi appears intact, while upon Etoposide and E64d+pepstatin treatment puncta start to form. How many Golgin97 puncta were identified in mock vs. treatment condition to begin with? Could the authors report % of puncta positive for Integrin $\alpha 5$ and Optn? How was the threshold for a 'positive signal' set? (same for panel 6i)
 - c. Why was a different type of quantification applied in panel 4e? Is there a morphological difference between intact Golgi and GOMED structures that could explain differences in localization?

3) Line 408 – YIPF3/4 are reported autophagy receptors localized to the Golgi apparatus.

Images in panel 8a are terrific.

Version 2:

Reviewer comments:

Reviewer #1

(Remarks to the Author)

In this additional round of revision, the authors have improved their manuscript and clarified my points raised previously.

Reviewer #2

(Remarks to the Author)

My concerns have been sufficiently addressed. I have the following remarks/questions to the revised manuscript:

Figure 6e – please adjust the y-axis to visualize individual data points and avoid that they are condensed in one spot.

Line 355 – please rephrase to reflect the sequence of events that is described in the study 1) ubiquitination, 2) binding of OPTN to ubiquitinated substrates

Line 363 – please rephrase: ‘These results indicate that integrin $\alpha 5$ may itself be polyubiquitinated...’ From the current data one cannot deduce whether ubiquitin is attached to integrin $\alpha 5$, to one of its interactors, or pulled down with OPTN (which likely also binds other K33-ubiquitinated substrates) in a complex.

Supplementary Figure 15 describes a new substrate MSLN. Could the authors indicate why this specific protein was chosen? The co-localization of MSLN and LAMP2 or OPTN upon etoposide and BafA1 treatment is not obvious. Could the authors comment on this in the text? Why was BafA1 used instead of E64d/pepstatin? BafA1 affects Golgi structure and this may impair analysis.

In analogy to integrin $\alpha 5$ - could the imaging data on MSLN be quantified?

REVIEWER COMMENTS

Reviewer #1 (Remarks to the Author):

The manuscript describes that ‘Optineurin is an adaptor protein for ubiquitinated substrates in Golgi membrane-associated degradation’. Golgi membrane-associated degradation (GOMED) is an autophagy independent processes that leads to the degradation of proteins that have passed through the trans-Golgi membranes upon stress. This manuscript aims to explore how GOMED substrates are selected for degradation. The authors use genetics, imaging and biochemical approaches to arrive at the conclusion that – in response to various stressors - Optineurin detects K33 polyubiquitylated proteins at the Golgi for GOMED. While this is an interesting and important concept, the manuscript in its current form, unfortunately falls short to provide compelling evidence to support the concepts. A major revision, substantial improvement of data quality and additional data would be required.

Response:

We greatly appreciate these constructive comments. In accordance with the comments, we extensively performed the following experiments: (1) analysis of the association between endogenous GOMED substrates and the OPTN-K33Ub pathway (Fig. 6), (2) analysis to determine if the increase in DAPGreen dots caused by the rescue of OPTN in PentaKO cells is owing to GOMED (Fig. 2–4), (3) confirmation that VSVG itself is a substrate of GOMED (Suppl. Fig. 7), and (4) collection of detailed data to confirm that the removal of mitochondria from immature mouse erythrocytes is mediated by GOMED. We also optimized some of the western blot analyses data. We believe that these experiments and explanations, as shown below, appropriately and sufficiently address your comments.

Major points:

- 1) What are the endogenous substrates of the OPTN-K33-GOMED pathway?
- 2) In many experiments it is difficult to directly discriminate GOMED and autophagy, due to the lack of an endogenous GOMED substrate.

Response:

We greatly appreciate this important question. In our original manuscript, we conducted experiments using artificial substrates for ease of analysis. However, as we have reported that integrin $\alpha 5$ is an endogenous substrate of GOMED (Ref. 7), we performed various experiments to elucidate whether integrin $\alpha 5$ is degraded by GOMED using the K33-OPTN pathway.

First, to clarify whether integrin $\alpha 5$ is a degradation substrate of autophagy, MEFs were

stimulated with rapamycin to induce autophagy, but the intracellular amount of integrin $\alpha 5$ was not changed (Suppl. Fig. 11a, b). In addition, we did not observe any colocalization between integrin $\alpha 5$ puncta and LC3 puncta (Suppl. Fig. 11a, d). In the case of etoposide-treated HeLa cells, the number of integrin $\alpha 5$ puncta was increased, but they were not colocalized with LC3 puncta (Fig. 6a, e), also showing that the degradation of integrin $\alpha 5$ was solely by GOMED. Moreover, we found that (1) the amount of integrin $\alpha 5$ increased when E64d + pepstatin A was administered (Fig. 6c), (2) integrin $\alpha 5$ was colocalized with a Golgi marker and OPTN (Fig. 6 f–i), (3) knockdown of OPTN increased the expression of integrin $\alpha 5$ (Fig. 6j), (4) integrin $\alpha 5$ binds to OPTN in a Ub-dependent manner (Fig. 6k), and (5) K33-Ub is utilized for this binding (Fig. 6l). The same results were also obtained in an experiment using HeLa cells treated with CBM (Suppl. Fig. 12). These data indicated that integrin $\alpha 5$ is degraded by GOMED via the K33Ub-OPTN pathway, and not by autophagy, suggesting that GOMED dynamics can be quantified by observing the degradation of integrin $\alpha 5$.

We added these results to the revised manuscript, and added the following description (page 14, line 1– page15, line 4).

“Endogenous GOMED substrate integrin $\alpha 5$ is also degraded via the K33-polyUb-OPTN pathway

We next investigated whether the same substrate recognition mechanism also functions for endogenous substrates. Integrin $\alpha 5$ is a molecule that is transported to focal adhesions via the Golgi apparatus, and we have previously reported that it is degraded by GOMED. First, we analyzed the relationship between integrin $\alpha 5$ and autophagy using immunostaining. Treatment with rapamycin, which induces autophagy alone, did not result in any changes in the expression levels (fluorescent intensity \times area/cell) of integrin $\alpha 5$ (Supplementary Fig. 11a, b). Furthermore, even when protein degradation was inhibited by the addition of E64d/pepstatin, the expression levels remained unchanged (Supplementary Fig. 11a, b). Although rapamycin increased the number of LC3 puncta (Supplementary Fig. 11a, c), LC3 puncta that colocalized with integrin $\alpha 5$ were scarcely observed (Supplementary Fig. 11a, d). These results suggest that integrin $\alpha 5$ is not a degradation substrate of autophagy⁷.

When etoposide is administered to HeLa cells, Golgi transport is first disrupted, causing integrin $\alpha 5$ to accumulate intracellularly (Fig. 6a “integrin $\alpha 5$ ”, c). At the same time, both autophagy and GOMED are induced (Fig. 2). Although integrin $\alpha 5$ well colocalize with Lamp1 (Fig. 6a “Merge”, d), they do not colocalize with LC3 (Fig. 6a, e), suggesting that a portion of the integrin $\alpha 5$ is degraded through GOMED. In fact, when protein degradation is inhibited by the addition of E64d/pepstatin, the expression level of integrin $\alpha 5$ increase (Fig. 6a, c). In contrast, in PentaKO cells, the integrin $\alpha 5$ fluorescence level was already increased with etoposide treatment

only, even without E64d/pepstatin, and no colocalization with LAMP1 was observed (Fig. 6b–e). Similar results were also obtained when cells were treated with CBM (Supplementary Fig. 12). In *Atg5^{KO}* MEFs, integrin $\alpha 5$ colocalized with OPTN on the Golgi (Fig. 6f, h) in a stimulus-dependent manner, and also with lysosomes (Fig. 6g, i). Furthermore, the expression level of integrin $\alpha 5$ increased upon the silencing of *Optn* (Fig. 6j). Importantly, integrin $\alpha 5$ interacted with Optn (Fig. 6k, lanes 5–7), and this interaction was substantially reduced by TAK-243 (Fig. 6k, lane 8). Furthermore, K33-Ub, but not K33R, was interacted with integrin $\alpha 5$ (Fig. 6l, lanes 14, 15), indicating that K33-polyubiquitinated integrin $\alpha 5$ interacts with OPTN, and is subsequently degraded by GOMED.”

- 3) The Data quality for some of the Western Blots experiments is suboptimal and (e.g.: Fig.3a, Fig.7c).

Response:

Thank you for your comments. We repeated several of the western blot experiments for accurate protein quantification (Fig. 1a, Fig. 3a, Fig. 5a, Fig. 8e [original Fig. 7c]).

- 4) Sometimes it is difficult to match the presented WB with the quantification (e.g. Fig. 1a, b). The WB clearly shows upregulation of p62 and NDP52 and OPTN protein levels (lanes 4,5,6), but in the quantification on OPTN is upregulated and p62 is even downregulated.

Response:

We appreciate this comment. We received the same comment from Reviewer#2 (Comment #5). In accordance with this comment, we showed more representative western blotting images, by which the discrepancy between the image data and quantitative data should be eliminated. (Fig. 1a, b, Fig. 3a, b).

- 5) The knock efficiency of OPTN appears to be rather moderate (Figure S2). OPTN knockdown experiments should be controlled by corresponding rescue experiments (eg. Figure 1c)

Response:

Thank you for your comment. To address this issue, we performed experiments using multiple siRNAs for *Optn*, and their rescue genes. First, we prepared four different siRNAs for Optn (no.1–4) and analyzed the knockdown efficiency using western blotting (Suppl. Fig. 2b). Next, we created siRNA-resistant Optn constructs (R1–4), transiently expressed them, and used them in the

rescue experiments. We confirmed the expression of each siRNA-resistant Optn by western blot (Suppl. Fig. 2b). Next, we conducted rescue experiments using the siRNA-resistant Optn constructs. We transfected *Atg5^{KO}* MEFs with siOptn, and subsequently expressed the siRNA-resistant Optn. These cells were stimulated with etoposide and analyzed with DAPGreen. Our results showed that the group in which Optn was rescued demonstrated a significant recovery of DAPGreen dots. This data was added as Suppl. Fig. 2.

We added these results to the revised manuscript, and added the following description (page 7, line 22– page 8, line 5).

“To strengthen this observation, we performed rescue experiments using four different siRNAs targeting Optn (siRNAs no.1–4) and their respective resistant gene created by substituting bases without altering the amino acid sequence (Flag-Optn R1–R4). As indicated, each siRNA achieved successful gene knockdown, and Optn expression (particularly 2 to 4) was well recovered by their respective resistant gene (Supplementary Fig. 2b). In these cells, the number of DAPGreen puncta induced by etoposide was substantially reduced by Optn siRNA, and recovered by their respective resistant plasmid (Supplementary Fig. 2c, d), confirming the important role of Optn as an adaptor protein in etoposide-induced GOMED. Therefore, the loss of Optn is expected to decrease the extent of GOMED by reducing the amount of recognized substrates.”

- 6) What is the cause for the difference in LC3 and DAPGreen punctae in p62 and NDP52 cells (Figure 2a,b,c)? does it imply that in addition to OPTN, also NBR1 and TXA1BP1 are GOMED receptors? In addition, the WB in Figure S4 (which should serve as a control for Figure 2a-c) lacks a critical control which makes it impossible to evaluate the protein expression levels of the adaptor proteins compared to the parental WT cells.

Response:

We appreciate this very important comment. We performed a newly designed experiment to address this comment. In the original manuscript, LC3 puncta and DAPGreen puncta appearing in etoposide-treated HeLa cells were observed in separate cells, so although the overall results were correct, in some cases (cells transfected with NBR1 or Tax1BP1, as you pointed out), more DAPGreen puncta were counted than LC3 puncta. Therefore, to more accurately quantify the results, we observed LC3 puncta and DAPGreen puncta in the same cells and measured them as DAPGreen⁺LC3⁺ puncta (autophagy) and DAPGreen⁺LC3⁻ puncta (GOMED). This enabled us to more clearly demonstrate that only OPTN can induce GOMED (Fig. 2a–c).

As you pointed out, Suppl. Fig. 3 (original Suppl. Fig. 4) lacked the proper control settings. We hence compared the endogenous adaptor proteins in WT HeLa cells with samples in which

the adaptor protein was exogenously expressed in PentaKO cells via western blot. The results confirmed the expression of the adaptor proteins in PentaKO cells, and its expression levels were not higher than those of the endogenous protein (Suppl. Fig. 3a and b).

We added these results to the revised manuscript, and added the following description (page 8, line 12– page 9, line 22).

“The crucial role of OPTN was also analyzed using HeLa cells and Pentaknockout (KO) HeLa cells lacking p62, NDP52, NBR1, Tax1BP1, and OPTN. For this purpose, we first expressed tagRFP-LC3 to these cells, treated them with etoposide, and then analyzed the formation of LC3 and DAPGreen puncta in the same cell. DAPGreen can recognize almost all autophagic structures from the phagophore to the autolysosome, as well as almost all GOMED structures (Supplementary Fig. 1d). Therefore, LC3-positive DAPGreen puncta are considered autophagic structures, while LC3-negative DAPGreen puncta are recognized as structures associated with GOMED. This has already been confirmed using the CLEM (Correlative Light and Electron Microscopy) assay (Supplementary Fig. 1). Additionally, a small number of LC3-positive but DAPGreen-negative puncta may appear, which are thought to represent non-autophagic LC3 puncta, such as those associated with LC3-associated phagocytosis (LAP). As a result of experiment, we found that both the number of DAPGreen⁺/LC3⁺ puncta (autophagic puncta) and the number of DAPGreen⁺/LC3⁻ puncta (GOMED puncta) were increased in HeLa cells, but not in PentaKO cells (Fig. 2a–c). Less GOMED occurring in PentaKO cells than control HeLa cells was confirmed by the absence of Ulk1⁷⁴⁶ phosphorylation signals, which is a specific marker of GOMED (Fig. 2d, e), suggesting that at least one of the autophagy adaptor proteins is involved in GOMED.

To identify molecule(s) involved in GOMED, we transfected plasmids encoding each of the adaptor proteins into PentaKO cells, and treated them with etoposide. Each protein was successfully expressed in PentaKO cells, with an expression level nearly the same as or less than the endogenous level in control HeLa cells (Supplementary Fig. 3a, b). Interestingly, the expression of each of p62, NDP52, NBR1, and TAX1BP1 restored the formation of DAPGreen⁺/LC3⁺ puncta, but not DAPGreen⁺/LC3⁻ puncta (Fig. 2a–c), indicating that all the adaptor proteins excluding OPTN play a role in etoposide-induced autophagy. In contrast, the expression of OPTN restored only DAPGreen⁺/LC3⁻ puncta (Fig. 2a, c), indicating that OPTN is involved in etoposide-induced GOMED in HeLa cells. The OPTN-dependent DAPGreen puncta were well merged with Rab9 (a crucial protein for GOMED) and GalT-mCherry (a Golgi marker) in etoposide-treated cells (Fig. 2f, Supplementary Fig. 4a–d), supporting that the DAPGreen puncta are GOMED structures. Some of the DAPGreen also colocalized with lysosomes, which is thought to represent autolysosome-like GOMED structures (Supplementary Fig. 4e, f). Regarding etoposide-induced autophagy, OPTN does not appear to be involved because LC3 puncta were not observed even after the

blockage of autophagy flux by E64d and pepstatin treatment, inhibitors of lysosomal protein degradation (Supplementary Fig. 5). All these data indicated that OPTN is involved in etoposide-induced GOMED, but not in autophagy in HeLa cells. The involvement of OPTN in GOMED is reasonable, because OPTN is known to localize to the Golgi, and GOMED membranes originate from Golgi membranes. “

- 7) Figure 2a: Does the DAPGreen in the OPTN rescued pentaKO cells co-localize with lysosomal markers?

Response:

We appreciate this comment. As you suggested, we analyzed whether the DAPGreen puncta that appear when etoposide is added to OPTN-expressing PentaKO cells colocalize with lysosomes (autolysosome-like GOMED structures) using LysoTracker staining. We found that many of the DAPGreen puncta colocalized with LysoTracker (Suppl. Fig. 4e, f). Some of the DAPGreen puncta were not colocalized with LysoTracker, but this was thought to be owing to the observation of autophagosome-like GOMED structures.

This point needed to be noted, so we have included it in the main text, as follows (page 9, lines 15–16).

“Some of the DAPGreen also colocalized with lysosomes, which is thought to represent autolysosome-like GOMED structures (Supplementary Fig. 4e, f).”

- 8) According to material and methods, the immunoprecipitation (IP) experiment used to show VSVG-GFP ubiquitylation were performed under ‘native’ conditions, such that interactions with other proteins would be retained. Hence proteins that interact with VSVG-GFP (or GFP fragments) could also be ubiquitylated. Moreover, the IP’s lack controls (e.g. GFP only and untagged VSVG) and the use of the E1 inhibitor is not sufficient.

Response:

Thank you for pointing this out. We received the same comments from Reviewer #2 (Comment #1). To approach this question, we performed two experiments. First, we performed immunoprecipitation of VSVG-GFP in a denatured condition, not just in a native condition. This confirmed that polyUb binds to OPTN even in a denatured condition (Suppl. Fig. 7a). Second, we investigated the presence or absence of polyubiquitination in VSVG only and GFP only. As a

result, we found that VSVG is indeed ubiquitinated and binds to OPTN, whereas GFP is not ubiquitinated (Suppl. Fig. 7b, c). We believe that these findings clearly confirm that VSVG is ubiquitinated and specifically binds to OPTN. We also added a sufficient amount of the E1 inhibitor in these experiments.

We added these results to the revised manuscript, and added the following description (page 11, lines 6–9 and page 11, lines 17–22)

“Ubiquitination was validated by the complete loss of PanUb signals using the specific E1 Ub activating enzyme inhibitor TAK-243 (Fig. 4a, lanes 6, 7), and by the presence of Ub signals even upon denaturing immunoprecipitation (Supplementary Fig. 7a, lane 8).”

“Consistently, when only VSVG was expressed, VSVG was ubiquitinated (Supplementary Figure 7b, lanes 10, 11) and Optn was coimmunoprecipitated with VSVG (Supplementary Figure 7b, lanes 4, 5), and these activities were abolished by TAK-243 (Supplementary Figure 7b, lanes 6, 12). In contrast, GFP on its own undergoes only minimal ubiquitination compared with VSVG-GFP, and no binding to Optn was observed (Supplementary Fig. 7c, lanes 5, 6).”

- 9) Based on the model propose VSVG-GFP should be degraded upon CBM treatment. Yet the fraction of VSVG-GFP that appears to be degraded is small (Figure 4e compare full length VSVG-GFP to the clipped/released GFP). Is the GFP release from full length VSVG-GFP dependent of the catabolic activity of lysosomes? Moreover, the bulk of VSVG-GFP accumulated at the Golgi (Figure 4d). How does that fit to the model?

Response:

Thank you very much for your helpful comment. As you have described, VSVG-GFP can be degraded by GOMED, and is detected as a GFP band. However, this GFP band is further degraded, so the actual amount of degradation is less than this. This point needed to be noted, so we have included it in the main text, as follows (page 12, lines 4–6).

“During GOMED, VSVG-GFP is first cleaved at the junction of the two proteins and subsequently the GFP part is degraded, so that a transient GFP cleavage band appears.”

Furthermore, it has been confirmed that the release of GFP from VSVG-GFP is dependent on lysosomal activity, as when lysosomal degradation is inhibited by adding chloroquine (CQ), VSVG-GFP accumulates and the amount of GFP release decreases (Suppl. Fig. 8). In addition, the colocalization of VSVG-GFP and LAMP1 was also observed upon immunostaining by CBM,

and which was substantially increased by CQ (Fig. 4f, g). We added these results to the revised manuscript, and added the following description (page 11, line 25–page 12, line 8).

“Furthermore, a proportion of VSVG-GFP was also colocalized with Lamp1 to be degraded in autolysosome-like GOMED structures in the cells expressing Optn (Fig. 4f, g). Note that this colocalization was not observed in the absence of Optn, and was increased by the addition of CQ (Fig. 4f, g). These data suggested that VSVG-GFP interacts with Optn in the Golgi, by which VSVG-GFP is delivered to autolysosome-like GOMED structures. Biochemical analysis confirmed this scenario. During GOMED, VSVG-GFP is first cleaved at the junction of the two proteins and subsequently the GFP part is degraded, so that a transient GFP cleavage band appears. Indeed, the cleaved GFP band increased time-dependently after CBM treatment, and was decreased by TAK-243 treatment (Fig. 4a–c). This cleavage was also suppressed by the addition of CQ (Supplementary Fig. 8).”

The results that VSVG-GFP is observed in the Golgi apparatus after CBM treatment is reasonable. This is because CBM is a drug that inhibits transport from the Golgi apparatus, so VSVG-GFP accumulates in the Golgi apparatus first. At the same time, GOMED is induced and some of it is degraded, but because not all of it is degraded, a signal on the Golgi apparatus is detected. In the revised manuscript, we replaced the VSVG-GFP/Golgi/Optn image (Figure 4d), aiming to show a more representative cell. We also added statistical analyses to clarify that VSVG-GFP and Optn are colocalized on the Golgi apparatus in a stimulus-dependent manner. We added these results to the revised manuscript, and added the following description (page 11, lines 23–25).

“Consistently, immunofluorescence analysis revealed that a proportion of VSVG-GFP was accumulated in the Golgi upon CBM treatment (owing to the blockage of trafficking from the Golgi to the plasma membrane) and was colocalized with Optn (Fig. 4d, e).”

10) Figure 5a: It seems that the expression of all Ub chains triggers VSVG-GFP degradation (except for K0 Ub, input lane 1)

Response:

We appreciate you pointing this out. This experiment involved the overexpression of Myc-tagged Ub chains and VSVG-GFP in HEK293T cells, and then treating them with CBM. Because the cells have endogenous Ub chains, the expression of the exogenous Ub chains should not have a substantial effect on the cleavage of VSVG-GFP. As you pointed out, our original manuscript did

not clearly show the cleavage of VSVG-GFP, so we optimized the experimental method and conducted the experiment again. As a result, we obtained the expected results with little effect of the Ub chain (Fig. 5a). These points are briefly described in the figure legend, as follows (page 43, lines 10–12).

“The cleavage of VSVG-GFP is not substantially affected by the type of exogenous Ub chains, owing to the presence of endogenous Ub.”

- 11) The ratio of full length VSVG-GFP to clipped GFP differs between experiments. What could be the reason for these substantial differences?

Response:

Thank you for pointing this out. This was simply a technical problem, so we adjusted the conditions and conducted the experiment again. As a result, we were able to obtain western blot data in which the appropriate amounts of VSVG-GFP and GFP are visible (Fig. 5a).

- 12) Figure S5: K63 polyubiquitin chains were also co-IP with OPTN -> these are the autophagy cargoes?

Results:

As you have pointed out, autophagy may be involved in this process. In this experiment (Figure S5 in the original manuscript; Figure S9 in the revised manuscript), HEK293T cells were treated with etoposide. At this time, autophagy is also activated in addition to GOMED. Therefore, it is highly likely that the K33Ub chains bind to OPTN as part of GOMED, and the K63Ub chains bind to OPTN as part of autophagy. Regarding the involvement of K63Ub chains during etoposide stimulation, we conducted experiments using K63R and confirmed that there was some involvement, albeit weak (Suppl. Fig. 9b). Therefore, although OPTN appears to be weakly involved in etoposide-induced autophagy (Fig. 2), it appears that K63 ubiquitination occurs to a small extent.

Furthermore, we also confirmed that K63 is not involved in the effects of CBM stimulation, which only occurs in GOMED (Suppl. Fig. 9c). These experimental results indicated that K33-Ub chains are important in GOMED, and that the involvement of the K63-Ub chains was minimal. However, we think that the involvement of K63-Ub chains in GOMED cannot be completely ruled out, so we have summarized these findings in the revised manuscript, as follows (page 13, lines 4–15).

“Similar to CBM treatment, when we expressed various types of Myc (3×)-tagged Ub chains together with Flag-OPTN into HEK293T cells, treated them with etoposide for 12 hr, and immunoprecipitated the lysates with Flag-OPTN, we found that K33-Ub strongly and K63-Ub weakly interacts with Flag-OPTN in these etoposide-treated cells (Supplementary Fig. 9a, lanes 15 and 17). Because the K63R signal appeared to be slightly weak (Supplementary Fig. 9b, lane 15), this suggests that in addition to K33-Ub, K63-Ub functions in GOMED upon etoposide stimulation. However, as etoposide stimulation induces both autophagy and GOMED, it is possible that K63 is only involved in autophagy. In fact, upon CBM stimulation, which induces only GOMED, there is no involvement of K63-Ub (the polyUb signal seen in K63R should indicate K33-Ub) (Supplementary Fig. 9c). Therefore, the involvement of K63-polyUb in GOMED is minimal, and K33 Ub chains are crucial for substrate recognition by OPTN during GOMED. “

13) The WB in Figure S6 is suboptimal – there are no bands visible for full length VSVG-GFP

Response:

We appreciate your comment. We have re-examined the western blot data and replaced the results with more appropriate data (Supplementary Fig. 10 in the revised manuscript).

14) Figure 7d, e: loss of OPTN could also cause mitophagy defects. It is impossible to discriminate the role of OPTN in autophagy or GOMED.

Response:

Thank you for your very important comments. We received the same comments from Reviewer #2 (Comment #3). In previous papers, we and other groups demonstrated that autophagy is not involved in the removal of mitochondria from immature red blood cells (Ref. 4 and 12). In fact, in *Atg5^{KO}* mice, red blood cell differentiation proceeds normally, including the removal of mitochondria. On the other hand, in mice lacking the *Ulk1* gene, which plays an important role in GOMED and autophagy, mitochondria remain within erythrocytes, and the erythrocytes become deformed. Thus, when comparing *Atg5^{KO}* mice and *Ulk1^{KO}* mice, it can be seen that autophagy is not involved in the removal of mitochondria from erythrocytes, and that GOMED is required for this process.

To demonstrate this phenomenon morphologically, we observed immature red blood cells from each mouse using electron microscopy. We found that in WT and *Atg5^{KO}* reticulocytes, mitochondria were engulfed into GOMED structures derived from the Golgi membrane, but

in *Ulk1*^{KO} reticulocytes, mitochondria were attached to the Golgi membrane but not engulfed (Fig. 8a). This confirms the involvement of GOMED in the removal of mitochondria. Furthermore, we isolated Ter119-positive cells (erythrocytes) from newborn mice and costained them for residual mitochondria and LC3. As a result, the colocalization of Tom20 (a mitochondrial marker) and Lamp2 was observed, but not the colocalization of LC3 (Fig. 8b). We believe these results confirm that mitochondrial removal in reticulocytes is specifically carried out by GOMED.

As you pointed out, OPTN has been confirmed to be involved in the removal of mitochondria by autophagy in neurons and other cells, but in red blood cells, this is not autophagy-mediated mitochondrial removal, so the abnormal removal of mitochondria and the abnormal morphology of red blood cells in *OPTN*^{KO} red blood cells are thought to be owing to the loss of substrate recognition by GOMED via OPTN. We added these results to the revised manuscript, and added the following description (page15, line 26– page16, line 12).

“Note that mitochondrial elimination occurs almost normally proceeds in *Atg5*^{KO} erythrocytes, whereas it is largely disrupted in erythrocytes lacking Ulk1, which is a key molecule both in autophagy and in GOMED^{1,4}, indicating that this phenomenon is GOMED-dependent. This is supported by morphological data, because condensed mitochondria were engulfed by Golgi-like membranes in erythroblasts from wild-type (WT) and *Atg5*^{KO} embryos (Fig. 8a, Supplementary Fig. 13). In contrast, in *Ulk1*^{KO} erythrocytes, in which GOMED is inhibited at an early step, multiple mitochondria were attached to the extended Golgi membrane (Fig. 8a, Supplementary Fig. 13). This morphology was owing to a dysfunction of the formation of phagophore-like GOMED structures. The GOMED-dependent elimination of mitochondria was also confirmed by the colocalization of Tom20 (a mitochondrial protein) with Lamp2, but not LC3, in Ter119⁺ cells (indicating erythrocytes) (Fig. 8b). These findings indicated that the removal of mitochondria from erythrocytes is carried out by GOMED.”

Minor points:

1: Figure 3C: The DAPI staining appears suboptimal.

Response:

Thank you very much for your comments. We have replaced the images with those in which DAPI staining is clearly visible (Fig. 3c).

2: Figure 3e: Please explain why LAMP2 is upregulated upon GOMED induction – are GOMED substrates sorted into these LAMP2 lysosomes?

Response:

Yes, large LAMP2 puncta were generated owing to the fusion of lysosomes with autophagosome-like GOMED structures, as described elsewhere^{6,39}. This was previously confirmed using correlative light and electron microscopy. Furthermore, in the case of CBM treatment, the LAMP2 expression level was also upregulated (Supplementary Fig. 6). We described the details of the LAMP2 assay, as follows (page10, lines 7– 17).

“GOMED was analyzed using LAMP2 immunostaining instead of DAPGreen, because LAMP2 signals appear as small puncta in lysosomes, but when they fuse with autophagosome-like GOMED structures (or autophagosomes) to form autolysosome-like GOMED structures (or autolysosomes), they can be observed as larger puncta (which are sometimes ring-shaped). The correspondence of large LAMP2 puncta to autolysosome-like GOMED structures (or autolysosomes) was previously confirmed using CLEM analysis^{6,42}. Therefore, this characteristic can be used as an indicator to evaluate the abundance of autolysosome-like GOMED structures (or autolysosomes). We found that many large LAMP2 puncta were generated upon CBM treatment in HeLa cells, whereas only a few were generated in PentaKO cells (Fig. 3e, g). Consistent results were obtained by LAMP2 immunoblot analysis (Supplementary Figure 6).”

Reviewer #2 (Remarks to the Author):

In this manuscript the authors set out to identify a substrate recognition mechanism of GOMED (Golgi membrane-associated degradation). GOMED is defined as a process that targets proteins that have passed through the Golgi for degradation (however, mitochondria are introduced as an additional substrate). As further stated in the manuscript, GOMED functions by generating autophagosome-like structures, whose formation does not depend on ATG9a, ATG5/7, or WIPI1/2 proteins.

Using VSVG-GFP, a previously established GOMED substrate the authors identify a role for Optineurin (OPTN), an autophagy receptor, as a substrate receptor for GOMED. They propose GOMED to be a ubiquitin-dependent pathway. Their data show that OPTN interacts with K33-linked Ub chains and that the ZF domain of OPTN is required for this interaction. The dependence of VSVG-GFP degradation (as reflected by the accumulation of free GFP) is dependent on OPTN in ATG5 KO cells. The authors further propose an OPTN-GOMED axis to be important for the degradation of mitochondria during erythroid maturation.

While this manuscript provides insights into the role of OPTN in ubiquitin-mediated VSVG-EGFP degradation and in the degradation of mitochondria, more evidence is needed to substantiate the proposed model linking GOMED, OPTN, and the two very distinct substrates

(VSVG-GFP, passing through the Golgi and mitochondria).

Response:

Thank you very much for your various constructive comments. In response to your comments, we performed various experiments. The main experiments are as follows.

- 1) We further analyzed and confirmed that VSVG is actually ubiquitinated (Supplementary Fig. 7).
- 2) We confirmed the colocalization of VSVG, OPTN, and Golgi upon CBM treatment (Fig. 4d, e).
- 3) We performed several erythrocyte experiments. Using EM, we confirmed that some mitochondria are engulfed by Golgi membranes in WT and *Atg5*^{KO} reticulocytes, but not in *Ulk1*^{KO} reticulocytes. Furthermore, mitochondria are not merged with LC3 in WT and *Atg5*^{KO} reticulocytes. These findings indicate that GOMED, but not autophagy, is involved in mitochondrial elimination during erythrocyte maturation (Fig. 8a, b, Supplementary Fig. 13).
- 4) We provided additional line of evidence from immunostaining that the DAPGreen puncta in PentaKO cells rescued by OPTN are autophagosome-like/autolysosome-like Golgi structures. We also added rescue experiments with OPTN to further demonstrate its necessity for the progression of GOMED (Fig. 2, Supplementary Fig. 10).

Major points:

- 1) The authors address the ubiquitination and degradation of the GOMED substrate VSVG-GFP (Figure 4). From the experiment shown in Figure 4a and b it cannot be concluded that VSVG is directly ubiquitinated. A denaturing IP needs to be performed to exclude that the poly-Ub signal stems from VSVG-GFP interaction partners. Does the VSVG-GFP construct contain lysine residues in its cytosolic tail?

Response:

Thank you very much for your comments. We received the same comments from Reviewer r#1 (Comment #8). To approach this question, we performed two experiments. First, we performed immunoprecipitation of VSVG-GFP in a denatured condition, and not just in the native condition. This confirmed that polyUb binds to OPTN even in a denatured condition (Supplementary Fig. 7a). Second, we investigated the presence or absence of polyubiquitination in VSVG only and GFP only. We found that VSVG is indeed ubiquitinated and binds to OPTN, whereas GFP is not ubiquitinated (Supplementary Fig. 7b, c). Furthermore, we found that the VSVG portion of

VSVG-GFP is located on the cytoplasmic side, and has several Lys residues (Ref. 45 and 46). These findings confirm that VSVG is ubiquitinated and specifically binds to OPTN. We added these results to the revised manuscript, and added the following description (page 11, lines 6–22).

“Ubiquitination was validated by the complete loss of PanUb signals using the specific E1 Ub activating enzyme inhibitor TAK-243⁴⁴ (Fig. 4a, lanes 6, 7), and by the presence of Ub signals even upon denaturing immunoprecipitation (Supplementary Fig. 7a, lane 8). Furthermore, when we analyzed whether Optn was coimmunoprecipitated with VSVG-GFP in CBM-treated *Atg5*^{KO} MEFs, as expected, Optn, but none of the other four adaptor proteins, was eluted as a VSVG-GFP-interacting protein (Fig. 4b, lane 5, Fig. 4c). The interaction with Optn was substantially inhibited by the addition of an E1 inhibitor (Fig. 4b, lane 6, Fig. 4c), and was not observed in untreated cells (Fig. 4b, lane 4, Fig. 4c), indicating that Optn interacts with ubiquitinated VSVG-GFP upon CBM treatment. Note that it has been reported that the VSVG portion of VSVG-GFP is facing the cytosolic side on the vesicle of the Golgi apparatus, the cytosolic side of VSVG contains several Lys residues^{45,46}, and it is suggested that this region is likely to undergo ubiquitination^{45,46}. Consistently, when only VSVG was expressed, VSVG was ubiquitinated (Supplementary Figure 7b, lanes 10, 11) and Optn was coimmunoprecipitated with VSVG (Supplementary Figure 7b, lanes 4, 5), and these activities were abolished by TAK-243 (Supplementary Figure 7b, lanes 6, 12). In contrast, GFP on its own undergoes only minimal ubiquitination compared with VSVG-GFP, and no binding to Optn was observed (Supplementary Fig. 7c, lanes 5, 6).”

Lines 195-201 – Minor: The cartoon in Figure 4g should be updated for the GFP domain facing the Golgi lumen.

Response:

Thank you very much for your important comment. Because both VSVG and GFP are reported to be facing to cytosol, we corrected the schematic diagram in Fig. 4j and Fig. 7e in the revised manuscript.

2) A quantification for the imaging data in Figure 4d should be included to assess the co-localization of OPTN and VSVG-GFP at the Golgi.

Response:

Thank you very much for pointing this out. In the original manuscript, only images were shown

regarding the colocalization of VSVG, OPTN, and the Golgi apparatus. Indeed, quantifiability is important, so in the revised manuscript, we have shown quantitative data and performed statistical analysis (Fig. 4d and e). As a result, we found that, as with the images, the colocalization of the VSVG, OPTN, and Golgi increased upon CBM treatment, indicating that CBM enhances the interaction of VSVG-GFP to OPTN on the Golgi. We added these results to the revised manuscript, and added the following description (page 11, lines 23–25).

“Consistently, immunofluorescence analysis revealed that a proportion of VSVG-GFP was accumulated in the Golgi upon CBM treatment (owing to the blockage of trafficking from the Golgi to the plasma membrane) and was colocalized with Optn (Fig. 4d, e).”

3) In the final Figure 7, the involvement of OPTN in the elimination of mitochondria from erythrocytes was addressed. This process has previously been shown to depend on Ulk1, an enzyme that is part of the ULK autophagy initiation complex. A knock-out of Ulk1 may impair GOMED (line 252) but in addition it also impairs classical autophagy and the accumulation of OPTN on mitochondria can therefore not be directly related to as dysfunction in GOMED. In general, OPTN is not an exclusive substrate receptor for GOMED but also interacts with LC3 proteins (LIR domain of OPTN) and functions in the classical autophagy pathway. OPTN has previously been shown to associate for example with damaged mitochondria and to target them for autophagic degradation. 10.1073/pnas.1405752111
The involvement of OPTN does not equal the involvement of GOMED. It can therefore not be assumed that phenotypes observed in OPTN KO cells are always related to the impairment of GOMED. Is there any evidence that Golgi membranes are forming autophagosome-like structures around mitochondria?

Response:

Thank you very much for your important comments. We received the same comments from Reviewer #1 (Comment #14). In the previous papers, we and other groups demonstrated that autophagy is not involved in the removal of mitochondria from immature red blood cells (Ref. 4 and 12). In fact, in *Atg5*^{KO} mice, red blood cell differentiation proceeds normally, including the removal of mitochondria. On the other hand, in mice lacking the *Ulk1* gene, which plays an important role in GOMED and autophagy, mitochondria remain within erythrocytes, and the erythrocytes become deformed. Thus, a comparison of *Atg5*^{KO} mice and *Ulk1*^{KO} mice demonstrates that autophagy is not involved in the removal of mitochondria from erythrocytes, and that GOMED is important.

As you suggested, to demonstrate this phenomenon morphologically, we observed immature red blood cells from each mouse using electron microscopy. We found that in WT and *Atg5*^{KO} reticulocytes, mitochondria were engulfed into GOMED structures derived from the Golgi membrane, but in *Ulk1*^{KO} reticulocytes, mitochondria were attached to the Golgi membrane but not engulfed (Fig. 8a). This confirms the involvement of GOMED in the removal of mitochondria. Furthermore, Ter119-positive cells (erythrocytes) were isolated from newborn mice and costained for residual mitochondria and LC3. Our results demonstrated the colocalization of Tom20 (a mitochondrial marker) and Lamp2, but not LC3 (Fig. 8b). These results confirm that mitochondrial removal in reticulocytes is specifically carried out by GOMED.

As you pointed out, OPTN has been confirmed to be involved in the removal of mitochondria by autophagy in neurons and other cells, but in red blood cells, autophagy-mediated mitochondrial removal does not occur. Therefore, the abnormal removal of mitochondria and the abnormal morphology of *OPTN*^{KO} red blood cells are thought to be owing to the loss of substrate recognition by GOMED via OPTN. We added these results to the revised manuscript, and added the following description (page15, line 26– page16, line 12).

“Note that mitochondrial elimination occurs almost normally proceeds in *Atg5*^{KO} erythrocytes, whereas it is largely disrupted in erythrocytes lacking Ulk1, which is a key molecule both in autophagy and in GOMED^{1,4}, indicating that this phenomenon is GOMED-dependent. This is supported by morphological data, because condensed mitochondria were engulfed by Golgi-like membranes in erythroblasts from wild-type (WT) and *Atg5*^{KO} embryos (Fig. 8a, Supplementary Fig. 13). In contrast, in *Ulk1*^{KO} erythrocytes, in which GOMED is inhibited at an early step, multiple mitochondria were attached to the extended Golgi membrane (Fig. 8a, Supplementary Fig. 13). This morphology was owing to a dysfunction of the formation of phagophore-like GOMED structures. The GOMED-dependent elimination of mitochondria was also confirmed by the colocalization of Tom20 (a mitochondrial protein) with Lamp2, but not LC3, in Ter119⁺ cells (indicating erythrocytes) (Fig. 8b). These findings indicated that the removal of mitochondria from erythrocytes is carried out by GOMED.”

- 4) Figure 7c – given the strong difference in ActB between WT and the Optn KO samples in conjunction with the weak signal for OPTN in WT-MEFs, it is difficult to assess any potential reduction in the OPTN levels/complete depletion of OPTN in No. 1-6.

Response:

Thank you very much for this important comment. In accordance with this comment, we reperformed additional western blot analyses and replaced the results with more appropriate data

(Fig. 8e of the revised manuscript).

- 5) Figure 1 – the quantification in panel b does not reflect the example image in panel a. Specifically the levels of OPTN strongly decrease (comparing 0 and 12h) based on the WB, but do not change according to the bar graph shown in b. Minor: The y-axis of all bar graphs is labeled as arbitrary units. Given that all zero-time point data points show 100 AU, the data are likely normalized and should be labelled as such.

Response:

Thank you very much for the comments. We received the same comments from Reviewer #1 (Comment #4). In accordance with these comments, we conducted additional experiments several times to improve the quality of the western blots, and showed representative western blotting images, in which the discrepancy between the imaging data and quantitative data should have been eliminated (Fig. 1a, b, Fig. 3a, b). Furthermore, as you pointed out, all zero-time point data plots have been normalized to 100 AU by which we were able to obtain more accurate results (Fig. 1b and 3b).

- 6) Related to (4) – Figure 3ab – which show similar data as in Figure 1ab with a different treatment – The levels of p62, Nbr1, and Ndp52 are barely detectable in the example images. The levels of TAX1BP1 also decrease upon CBM treatment in ATG5 KO cells. Can the authors comment on the potential involvement of TAX1BP1?

Response:

Thank you very much for your suggestion. As you pointed out, in the original manuscript, the expression level of Tax1Bp1 upon CBM treatment did not change in the quantitative data, but it appeared to decrease in the images (original Fig. 3a). Therefore, we selected more representative images (revised Fig. 3a). We also reperformed the experiment, and confirmed that, as in the quantitative data, there was no change in the expression level of Tax1Bp1.

- 7) The authors define DAPGreen+/LC3- puncta as GOMED puncta. The co-localization of these puncta with a GOMED marker needs to be included. The LC3- negative puncta may be positive for another mammalian ATG8-type protein and relate to classical autophagy pathways. Structures defined as GOMED puncta need to be clearly distinguished from classical autophagosomes.

Response:

Thank you very much for this comment. To define DAPGreen⁺/LC3⁻ dots as GOMED dots, we performed simultaneous staining with GOMED-associated molecules, such as Rab9 and the *trans*-Golgi marker (GalT-mCherry) which is a source membrane of GOMED. As a result, DAPGreen puncta in PentaKO cells rescued by OPTN colocalized with Rab9 and GalT-mCherry, confirming that these puncta are identical to GOMED puncta. These data are shown in Supplementary Fig. 4a-d of the revised manuscript. We described this point in the revised manuscript, as follows (page 9, lines 12– 15)

“The OPTN-dependent DAPGreen puncta were well merged with Rab9 (a crucial protein for GOMED) and GalT-mCherry (a Golgi marker) in etoposide-treated cells (Fig. 2f, Supplementary Fig. 4a–d), supporting that the DAPGreen puncta are GOMED structures.”

Minor points:

① The introduction falls a bit short in its description of autophagy and ubiquitin signaling.

For example: ...autophagy mainly degrades cytosolic proteins, whereas GOMED mainly degrades molecules that have passed through the Golgi. – The substrate spectrum of autophagy is much broader (including cellular organelles (mitochondria, lysosomes, etc), intracellular pathogenic organisms) and should also be reported as such.

The Ub code is more complex than reported in lines 80-83. More than K48, K63 and linear ubiquitin chains were shown to induce proteasomal degradation or autophagy (lines 93-95). Given the topic and conclusion of the paper, please add a more detailed introduction on the role/function of different Ub chains.

Response:

In accordance with this comment, we added more detailed explanations of autophagy and Ub to the revised manuscript, as follows (page 4, lines 4–25, and page 5, lines 4–20)

“The role of autophagy is to degrade unnecessary and harmful intracellular substances (such as damaged lysosomes, mitochondria, abnormal proteins, and intracellular parasites), and to maintain cellular homeostasis. Initially, autophagy was considered to be a non-selective degradation system that randomly engulfs cytoplasmic components. However, when degrading unnecessary and harmful substances, autophagy can selectively engulf them, in a process known as “selective autophagy”. GOMED is another intracellular proteolysis mechanism, which mainly degrades

proteins that pass through the Golgi apparatus, such as secreted proteins and plasma membrane proteins^{4,5,6}. GOMED is morphologically similar to autophagy, as both GOMED and autophagy engulf substrates in double-membrane structures, namely autophagosomes, and degrade them by fusion with lysosomes^{4,6}. These two proteolysis mechanisms are also similar in that they are phylogenetically conserved from yeast to mammals⁶. However, they operate in different contexts, have different cellular functions, and degrade different cargos^{6,7}. Furthermore, autophagic membranes originate from the endoplasmic reticulum (ER) membrane, particularly mitochondria-associated ER membranes^{8,9}, whereas GOMED membranes are derived from *trans*-Golgi membranes^{4,6}. Regarding the molecules involved, autophagy is driven by functional complexes containing Atg9, Wipi1/2, the Atg5-conjugation system, and the LC3-conjugation system^{1,10}, but none of these molecules are required for GOMED⁴. GOMED requires different molecules, namely, Wipi3/4 and Rab9, which have minimal function in autophagy^{4,11}. Regarding their physiological functions, GOMED functions in cell maintenance via its degradation of ceruloplasmin in neurons, and in mitochondrial removal in the final step of mitochondrial degradation in erythrocytes^{11,12,13}, but autophagy is not involved in these phenomena.”

“Ubiquitination is one of the most common post-translational modifications in a cell, and Ub chains can be linked via any of its seven Lys residues (K6, K11, K27, K29, K33, K48, and K63) or via the amino terminal M1 residue (generating a linear chain)^{16,17,18}. It is well known that polyUb bound to K48 acts as a signal for protein degradation by the proteasome, and that polyUb bound to K63 acts as a signal for protein degradation by autophagy^{19,20,21,22}. In addition to the major Ub chains, such as K48, K63, and M1 that have been analyzed to date, progress has been made recently in the analysis of atypical Ub chains, which are involved in specific intracellular processes and signal transduction pathways with diverse functions. For example, K6-linked chains are essential for DNA repair²³, whereas K11-linked chains regulate the cell cycle, particularly during the progression and termination of mitosis²⁴, and control protein degradation during cell division. K27- and K29-linked chains are involved in immune responses and inflammatory signaling, with K29-linked chains specifically aiding in the removal of abnormal proteins, such as Mitochondrial antiviral signaling, to support immune system function^{25,26}. Additionally, K33-linked chains contribute to intracellular transport and signal transduction regulation by ensuring the proper localization of specific proteins²⁷. Together, these atypical Ub chains are crucial for maintaining cellular homeostasis and facilitating adaptive responses.”

② How were the time points chosen for experiments shown in 1e and g? Based on WB in panel a, OPTN is largely degraded at the 12h time point. A time course experiment or additional analysis

of an earlier time point would strengthen the argument that OPTN localizes to Golgi membranes upon etoposide treatment.

Response:

Thank you for pointing this out. As GOMED has sufficiently advanced by 12 hr after etoposide stimulation, the Golgi localization of OPTN is thought to precede this. For this reason, we have added measurements at the 4-hr point. As a result, we were able to confirm that some OPTN, albeit a small amount, is localized to the Golgi apparatus at the 4-hr timepoint; i.e., it was upregulated and colocalized with the Golgi apparatus (Fig. 1e–h).

③ Can the authors comment on to why GFP is not degraded by GOMED, but remains as a degradation product from VSVG-GFP degradation in lysosomes?

Response:

Thank you very much for your helpful comment. We received the same comments from Reviewer #1 (Comment #9). During GOMED, VSVG-GFP is first cleaved at the junction between the two proteins, and subsequently the GFP part is degraded, so that a transient GFP cleavage band appears. We described this point in the revised manuscript as follows. (page 12, lines 4–9).

“During GOMED, VSVG-GFP is first cleaved at the junction of the two proteins and subsequently the GFP part is degraded, so that a transient GFP cleavage band appears. Indeed, the cleaved GFP band increased time-dependently after CBM treatment, and was decreased by TAK-243 treatment (Fig. 4a–c). This cleavage was also suppressed by the addition of CQ (Supplementary Fig. 8). Importantly, these cleaved bands were substantially reduced by the silencing of *Optn* (Fig. 4h, i).”

④ The presentation of the experiments performed with Ubiquitin mutants lacks clarity. From the text on page 8, it is not clear which type of Ub mutants were used – presumably Ubiquitin which exclusively harbors one lysine residue – K33? Please clarify.

Response:

Thank you very much for this comment. To clarify the definition of the Ub chain abbreviations, we added the following explanation to the revised manuscript, as follows (page 12, lines 15–20).

“we expressed various types of Myc (3 \times)-tagged Ub chains and VSVG-GFP into HEK293T cells, treated them with CBM for 12 hr, and then analyzed the Ub chains interacting with VSVG-GFP. Ub has seven Lysine residues, and the Ub in which all these Lys residues are mutated to Arginine (Arg) is defined as K0. Additionally, Ub in which only the x-th Lys residue remains is referred to as Kx. Furthermore, Ub in which only the x-th Lys is mutated to Arg is referred to as KxR.”

Point-by-point responses to the Reviewers' comments

Reviewer #1 (Remarks to the Author):

In my view, the revised manuscript unfortunately does not to provide compelling evidence that endogenous Golgi substrates for GOMED are K33-ubiquitinated and subsequently targeted into lysosomes by Optn. Specifically, the experiments using integrin $\alpha 5$ as an endogenous substrate (presented in Figures S11 and 6) for GOMED are not particularly convincing for the following reasons:

Response:

Thank you very much for your valuable comments. We have improved the accuracy of the data regarding integrin $\alpha 5$ as an endogenous substrate of GOMED, and additionally included new data on mesothelin, which is another endogenous GOMED substrate (re-revised manuscript, Suppl. Fig. 15). For both proteins, we demonstrated that they are recognized by OPTN and then ubiquitinated, and are subsequently degraded via the GOMED pathway. Furthermore, we added data not only of HeLa cells and PentaKO cells, but also of HexaKO (autophagy-deficient) cells, and HeLa cells incubated with GSK872, in which GOMED is suppressed. In HexaKO cells, integrin $\alpha 5$ was degraded upon etoposide treatment, whereas in HeLa cells incubated with GSK872, integrin $\alpha 5$ was not degraded and its cellular levels increased. Integrin $\alpha 5$ did not colocalize with LC3 or LAMP2 in GSK872-treated cells (re-revised manuscript, Fig. 6). These data indicate that integrin $\alpha 5$ is degraded by GOMED.

1. Immunofluorescence (IF) staining alone does not provide sufficient evidence to draw conclusions regarding the protein expression levels of integrin $\alpha 5$.

Response:

Thank you for your insightful comment. We agree that immunostaining data alone is insufficient to draw definite conclusions regarding the expression level of integrin $\alpha 5$. We have hence added western blot data and confirmed that they are consistent with the immunostaining results (re-revised manuscript, Fig. 6a and b).

2. There is clear co-localization of integrin $\alpha 5$ with LC3. Of course this is less pronounced compared to LAMP1, due to the relatively lower abundance of LC3 dots.

Response:

As you pointed out, a small number of puncta showing colocalization between integrin $\alpha 5$ and

LC3 were observed. However, the number was much less than that of puncta showing colocalization between integrin $\alpha 5$ and Lamp2 (re-revised manuscript, Fig. 6d and e).

Furthermore, we believe that integrin $\alpha 5$ -LC3 double-positive puncta are not generated by canonical autophagy, but rather by the fusion between canonical autolysosomes and autolysosome-like structures generated via GOMED. To confirm this, as our first approach, we knocked down syntaxin 17 (STX17). As STX17 is required for the fusion of canonical autophagosomes with lysosomes, its knockdown prevents autolysosome formation and thereby blocks the fusion between GOMED-derived structures and canonical autolysosomes. As a result of this treatment, integrin $\alpha 5$ -LC3 double-positive puncta, which had been observed occasionally, were no longer detected. We showed this data as Suppl. Fig. 13 of the re-revised manuscript.

As a second approach, we administered GSK872, a RIP3 inhibitor known to suppress etoposide-induced GOMED, to HeLa cells. As a result, the numerous integrin $\alpha 5$ -Lamp2 puncta observed in etoposide-treated HeLa cells were substantially reduced. Moreover, if integrin $\alpha 5$ was degraded via canonical autophagy, inhibition of GOMED would be expected to increase the number of integrin $\alpha 5$ -LC3 double-positive puncta. However, no such increase was observed (re-revised manuscript, Fig. 6c and e). Furthermore, the intracellular levels of integrin $\alpha 5$ were increased upon GSK872 treatment, and this increase was not further augmented by E64d treatment (re-revised manuscript, Fig. 6a and b), indicating that integrin $\alpha 5$ is indeed a substrate of GOMED.

Additionally, the frequent colocalization of integrin $\alpha 5$ with Rab9 (a marker of GOMED) further supports this conclusion (re-revised manuscript, Suppl. Fig. 14a). In contrast to HeLa cells incubated with GSK872, in HexaKO cells, in which all LC3 family members are deleted and canonical autophagy is abolished, the expression level of integrin $\alpha 5$ remains low (re-revised manuscript; Fig. 6a, b). Furthermore, treatment of these cells with E64d/pepstatin increased the expression level of integrin $\alpha 5$ (re-revised manuscript, Fig. 6a and b), demonstrating that it is degraded even in the absence of autophagy. All of these results indicate that integrin $\alpha 5$ is not degraded via canonical autophagy, but is degraded by GOMED.

3. Curiously, Optn appears to constitutively interact with integrin $\alpha 5$ even in the absence of etoposide treatment. Furthermore, this interaction seems to be largely independent of ubiquitination, as it persists - albeit at a slightly reduced level - even after E1 inhibition, which effectively abrogates integrin $\alpha 5$ ubiquitination.

Response:

Thank you for your valuable comment. As integrin $\alpha 5$ is constitutively recycled by GOMED, it is likely that some interaction with OPTN occurs even under untreated conditions. The relatively weak effect of the E1 inhibitor is likely owing to the short treatment duration (only 1 hr). In fact, when the treatment time was extended (12 hr), ubiquitin bands were almost completely abolished, and the interaction between integrin $\alpha 5$ and OPTN was no longer observed. These data have been included in Fig. 6j of the re-revised manuscript as a replacement for the previous data (Fig. 6k of the previous revised manuscript).

Reviewer #2 (Remarks to the Author):

In the revised version the authors present additional data to substantiate their findings.

The following concerns regarding the data and conclusions remain:

1) Data on integrin $\alpha 5$ was added to analyze the substrate-targeting mechanism of GOMED i.e. OPTN in the context of an endogenous substrate.

Response:

Thank you very much for your constructive comments. To enhance the credibility of our findings regarding the endogenous substrate, we have added the following data. First, we analyzed the expression levels of integrin $\alpha 5$ in HeLa cells by western blotting (re-revised manuscript, Fig. 6a and b). Furthermore, we analyzed HexaKO cells, which lack all LC3 family genes and thus have no canonical autophagy activity, and HeLa cells incubated with GSK872, in which GOMED is suppressed. In HexaKO cells, the expression level of integrin $\alpha 5$ was generally low. However, it increased upon etoposide treatment and was further increased by the addition of E64d/pepstatin, suggesting that canonical autophagy is unlikely to be involved in its degradation. In contrast, in HeLa cells treated with GSK872, integrin $\alpha 5$ levels were increased even at the basal level, and additional treatment with E64d/pepstatin had little effect (re-revised manuscript, Fig. 6a and b). Furthermore, integrin $\alpha 5$ did not colocalize with LC3 and LAMP2 in GSK872-treated cells. These data indicate that integrin $\alpha 5$ is not degraded by autophagy, but rather by GOMED (re-revised manuscript, Fig. 6). We also included new data on mesothelin, which is another endogenous GOMED substrate (re-revised manuscript, Suppl. Fig. 15).

a. The authors refer to changes in integrin expression levels – is there evidence that the expression of integrin $\alpha 5$ changes? If not, please remove the term ‘expression’.

Response:

Thank you very much for the important comments. We received a similar comment from Reviewer #1. Accordingly, we analyzed the expression levels of integrin $\alpha 5$ in HeLa cells by western blotting. In addition, we deliberately used the term “expression” for the western blot data and “fluorescence signal” for the immunostaining data, to distinguish between the two types of measurements in the re-revised manuscript.

b. Panel c suggests that the levels of integrin $\alpha 5$ change upon etoposide treatment. In panel k however, the Western blot data show that the levels of integrin $\alpha 5$ do not change. How can these data be reconciled? Could the effect observed in panel a be due to changes in localization/clustering rather than a change in levels?

Response:

Thank you very much for your insightful comments. In accordance with the comments, we conducted a more thorough set of experiments in this section. As you rightly pointed out, in the immunofluorescence data presented in Fig. 6a and 6c of the previous revised manuscript, the fluorescence intensity increased in a stimulus-dependent manner. In contrast, the western blot data shown in Fig. 6k of the previous manuscript appeared to show minimal changes in band intensity. However, as the original western blot results were relatively faint and lacked quantitative analysis, we repeated the western blot experiments and performed densitometric quantification. As a result, we were able to confirm that there is no inconsistency between the fluorescence intensity data obtained from immunostaining (re-revised manuscript, Fig. 6c and f) and the western blot images and their quantification (re-revised manuscript, Fig. 6a, b, i, and j). Although it is true that fluorescence-based measurements cannot provide precisely quantify protein levels, owing to factors such as the subcellular localization of proteins, the western blot data complement and support the immunofluorescence results. Therefore, taken together, we believe that our results on the changes in integrin $\alpha 5$ expression in response to GOMED stimulation are both consistent and reliable.

c. Why was the fluorescence intensity x area/cell quantified (assuming mean fluorescence intensity?) rather than the integrated intensity per cell (panel 6c and SI)?

Response:

In accordance with your suggestion, we standardized the quantification method for all similar imaging data, using integrated intensity per cell (re-revised manuscript, Fig. 6f, Suppl. Fig. 11b and 12b).

d. The description in lines 319-312 does not fully match the data shown in Fig 6c. Based on the quantification the levels of integrin $\alpha 5$ increase in the PentaKO cells without any treatment. The relative increase produced by Etoposide and E64d+pepstatin treatment resembles the effect observed in WT cells.

Response:

Thank you very much for your comment. In the previous manuscript, we stated, based on the data in Fig. 6c, that “in PentaKO cells, the integrin $\alpha 5$ fluorescence level was already increased with etoposide treatment only, even without E64d/pepstatin, and no colocalization with LAMP1 was observed.” However, as you correctly pointed out, integrin $\alpha 5$ levels were already increased in untreated PentaKO cells, and this point was not adequately described in the previous text. Therefore, in the re-revised manuscript, we have clarified this by accurately stating, based on the data shown in Fig. 6a and 6b (re-revised manuscript), as follows.

“integrin $\alpha 5$ levels in PentaKO cells were high even at the basal level, and did not increase any further upon etoposide and E64d/pepstatin treatment (Fig. 6a, b).”

e. Panel j represents a strong result for Optn involvement in integrin $\alpha 5$ turnover (while this does not equal an involvement of GOMED).

Response:

Thank you very much for this valuable comment. As you pointed out, Fig. 6i of the re-revised manuscript (Fig. 6J of the previous manuscript) indicates the involvement of OPTN but does not directly demonstrate the involvement of GOMED. However, in the re-revised manuscript, we believe that the degradation of integrin $\alpha 5$ via GOMED is sufficiently demonstrated in Fig. 6a–f. Therefore, we considered that the purpose of this figure (Fig. 6i) is primarily serve to show the involvement of OPTN in integrin $\alpha 5$ degradation. With this point in mind, we revised the description as follows.

“In addition to the colocalization of integrin and OPTN, knockdown of *Optn* led to an increase in intracellular integrin $\alpha 5$ levels, strongly suggesting that OPTN is involved in the GOMED-mediated degradation of integrin $\alpha 5$ (Fig. 6i).”

2) How were the imaging data quantified? I could not identify a detailed methods section on this aspect.

Response:

We appreciate your insightful comment. As the description of the image data analysis was incomplete, we have provided additional details as follows. Moreover, we have re-analyzed several immunofluorescence image datasets as described below.

“Triple-color immunofluorescence images were analyzed using ImageJ software (Fuji). Specifically, fluorescence channels were split and binarized by manual thresholding to isolate specific signals. Logical AND operations were applied using the “Image Calculator” function to identify regions positive for all three markers. The area of triple-colocalized regions was quantified using the “Analyze Particles” function, and results were obtained from the summary output. Integrated fluorescence intensities were measured using ImageJ, by defining regions of interest (ROIs) and using the “Measure” function to obtain integrated densities. The background signal was determined using regions adjacent to the ROIs, and subtracted from the data. Colocalization between two fluorescence signals was quantified using the Pearson’s correlation coefficient in ImageJ. After channel splitting and background subtraction, the “Coloc 2” plugin was used to calculate Pearson’s correlation coefficients within defined ROIs. Results were based on at least 10 cells per condition.”

a. In Fig 1f and h, puncta were quantified, though I could not identify any puncta in the images.

Response:

We agree that a substantial proportion of the signals in Fig. 1f and 1h cannot be clearly identified as puncta. Accordingly, the data were reanalyzed using Pearson’s correlation coefficient, which is an appropriate measure for evaluating colocalization.

b. In Fig 6h Golgin97 puncta were quantified – In mock-treated cells, the Golgi appears intact, while upon Etoposide and E64d+pepstatin treatment puncta start to form. How many Golgin97 puncta were identified in mock vs. treatment condition to begin with? Could the authors report % of puncta positive for Integrin $\alpha 5$ and Optn? How was the threshold for a ‘positive signal’ set? (same for panel 6i)

Response:

As with our response to the previous comment 2a, a substantial proportion of the signals shown in Fig. 6g and h of the re-revised manuscript (same as Fig. 6f and h of the previous manuscript) could not be clearly identified as puncta. To address this point, we reanalyzed the triple immunofluorescence staining data and quantified the area of colocalization among all three signals, namely, Golgin97, integrin $\alpha 5$, and OPTN, using ImageJ software. Specifically, we used the “Colocalization” function in the Image Calculator and analyzed the overlapping regions where all three fluorescence signals were present. To ensure accurate quantification, we first applied individual thresholds for each fluorescence channel based on background-corrected intensity histograms, thereby defining a “positive signal” for each marker. Triple-positive puncta were identified as regions exceeding the threshold in all three channels.

c. Why was a different type of quantification applied in panel 4e? Is there a morphological difference between intact Golgi and GOMED structures that could explain differences in localization?

Response:

As described above, we analyzed the image shown in Fig. 6g using the same method as that used in Fig. 4d and e.

3) Line 408 – YIPF3/4 are reported autophagy receptors localized to the Golgi apparatus.

Response:

Thank you very much for this comment. in accordance with the comment, we revised the sentence as follows.

“It is reasonable that OPTN, one of the five autophagy adaptor molecules reported to localize to the Golgi, is involved in GOMED, because the Golgi membrane is the membrane source of GOMED^{4,6}. Recent studies have identified autophagy adaptors specific to Golgiphagy^{57,58}; however, in the context of GOMED, OPTN acts as a key mediator.”

Images in panel 8a are terrific.

Response:

We sincerely appreciate your positive evaluation.

Point-by-point responses to the Reviewers' comments

Reviewer #1 (Remarks to the Author):

In this additional round of revision, the authors have improved their manuscript and clarified my points raised previously.

Response:

We sincerely appreciate your positive evaluation and your constructive feedback throughout the review process, which have greatly helped us to improve the manuscript.

Reviewer #2 (Remarks to the Author):

My concerns have been sufficiently addressed.

Response:

We are sincerely grateful for your positive evaluation and constructive feedback, which have been invaluable in helping us to further improve the manuscript.

I have the following remarks/questions to the revised manuscript:

Response:

As you rightly noted, we have made several modifications to the Figures and revised relevant parts of the text accordingly. Below, we provide detailed responses to each of your remarks and questions.

1, Figure 6e – please adjust the y-axis to visualize individual data points and avoid that they are condensed in one spot.

Response:

We appreciate your suggestion. In response, we have adjusted the y-axis to allow clear visualization of individual data points and to avoid their condensation in a single spot (Figure 6e).

**2, Line 355 – please rephrase to reflect the sequence of events that is described in the study
1) ubiquitination, 2) binding of OPTN to ubiquitinated substrates**

Response:

We appreciate this suggestion and have revised the text to reflect it accordingly.

Original :

We next investigated whether the endogenous GOMED substrate integrin $\alpha 5$ is recognized by OPTN and then polyubiquitinated via K33-linked ubiquitin chains.

Revised :

We next investigated whether the endogenous GOMED substrate integrin $\alpha 5$ is polyubiquitinated via K33-linked ubiquitin chains and then recognized by OPTN (Line 354).

2, Line 363 – please rephrase: ‘These results indicate that integrin $\alpha 5$ may itself be polyubiquitinated... ‘ From the current data one cannot deduce whether ubiquitin is attached to integrin $\alpha 5$, to one of its interactors, or pulled down with OPTN (which likely also binds other K33-ubiquitinated substrates) in a complex.

Response:

We are grateful for your constructive comments. To ensure consistency with the experimental results, we have revised the sentence as follows:

Original :

These results indicate that integrin $\alpha 5$ is polyubiquitinated via K33 linkages, recognized by OPTN, and subsequently degraded through GOMED.

Revised :

These results indicate that OPTN and K33-linked polyubiquitin are involved in the GOMED-mediated degradation pathway of integrin $\alpha 5$ (Line 363).

3, Supplementary Figure 15 describes a new substrate MSLN. Could the authors indicate why this specific protein was chosen?

Response:

Thank you for your question. We performed a comprehensive mass spectrometry analysis to identify molecules degraded in response to various GOMED stimuli. While we did not include all of these data in the current study, we selected several candidate molecules with potential

ubiquitination sites for further analysis. Among these, we focused on MSLN as a representative substrate of GOMED-mediated degradation.

4, The co-localization of MSLN and LAMP2 or OPTN upon etoposide and BafA1 treatment is not obvious. Could the authors comment on this in the text?

Response:

We are grateful for your insightful suggestion. Our analysis of the imaging data demonstrated that, although etoposide stimulation induced colocalization of MSLN with LAMP2 and OPTN, the degree of colocalization was relatively modest compared with that observed for substrates such as integrin $\alpha 5$ and VSVG. One possible explanation is that the intensity and duration of etoposide stimulation may not represent optimal conditions for accurately detecting the colocalization of MSLN with LAMP2 or OPTN.

Accordingly, we have added the following sentence to the main text (Line 373):

“Etoposide stimulation induced colocalization of MSLN with LAMP2 and OPTN, but to a lesser extent than with substrates such as integrin $\alpha 5$ and VSVG, possibly due to suboptimal stimulation conditions.”

5, Why was BafA1 used instead of E64d/pepstatin? BafA1 affects Golgi structure and this may impair analysis.

Response:

We appreciate your insightful comment. For the analysis of GOMED, we chose BafA1 to inhibit the degradation stage, which allowed us to assess pathway flux rather than only steady-state levels. Another important reason for using BafA1 is its ability to block autophagosome–lysosome fusion, thereby preventing canonical autolysosomes from merging with those derived from the GOMED pathway during LC3–MSLN colocalization analysis. In contrast, E64d and pepstatin inhibit lysosomal degradation but still permit fusion, which would have obscured the distinction between these compartments.

As you rightly noted, BafA1 can affect Golgi morphology, and such changes may have had some impact on our analysis. However, we believe these effects were minimal and unlikely to alter the main interpretation of our findings.

6, In analogy to integrin $\alpha 5$ - could the imaging data on MSLN be quantified?

Response:

We appreciate your valuable comment. In response, we performed a statistical analysis of the imaging data on MSLN and incorporated the results into the Supplemental Figure 15d, f, h, j.